# Wide-temperature-range thermoelectric n-type Mg$_3$(Sb,Bi)$_2$ with high average and peak $zT$ values

Jing-Wei Li[1], Zhijia Han[2], Jincheng Yu [1], Hua-Lu Zhuang [1] ✉, Haihua Hu[1], Bin Su [1], Hezhang Li[1], Yilin Jiang [1], Lu Chen[1], Weishu Liu [2], Qiang Zheng [3] & Jing-Feng Li [1] ✉

Mg$_3$(Sb,Bi)$_2$ is a promising thermoelectric material suited for electronic cooling, but there is still room to optimize its low-temperature performance. This work realizes >200% enhancement in room-temperature $zT$ by incorporating metallic inclusions (Nb or Ta) into the Mg$_3$(Sb,Bi)$_2$-based matrix. The electrical conductivity is boosted in the range of 300–450 K, whereas the corresponding Seebeck coefficients remain unchanged, leading to an exceptionally high room-temperature power factor >30 μW cm$^{-1}$ K$^{-2}$; such an unusual effect originates mainly from the modified interfacial barriers. The reduced interfacial barriers are conducive to carrier transport at low and high temperatures. Furthermore, benefiting from the reduced lattice thermal conductivity, a record-high average $zT$ > 1.5 and a maximum $zT$ of 2.04 at 798 K are achieved, resulting in a high thermoelectric conversion efficiency of 15%. This work demonstrates an efficient nanocomposite strategy to enhance the wide-temperature-range thermoelectric performance of n-type Mg$_3$(Sb,Bi)$_2$, broadening their potential for practical applications.

Thermoelectric (TE) technology that enables the direct conversion between heat and electricity is a promising approach to easing the fossil fuel shortage and promoting the sustainable development[1,2]. The energy conversion efficiency of TE devices is determined by the performance of TE materials, which is characterized by the dimensionless figure of merit, $zT = S^2\sigma T/(\kappa_L + \kappa_e)$, where $S$ is the Seebeck coefficient, $\sigma$ is the electrical conductivity, $T$ is the absolute temperature, $\kappa_L$ is the lattice thermal conductivity, and $\kappa_e$ is the electronic thermal conductivity. Considering that $S$, $\sigma$, and $\kappa_e$ are strongly coupled with each other through the carrier concentration, the key to enhancing the TE performance lies in decoupling or synergistically controlling these TE parameters.

Most TE materials, such as PbTe[3,4], SnSe[5,6], and Cu$_2$Se[7,8], exhibit superior performance at high temperatures (approximately above 773 K), whereas their room-temperature performance is inferior to the commercial Bi$_2$Te$_3$[9,10]. However, due to the high cost of Te element, it is vital to seek and develop economical substitutes for Bi$_2$Te$_3$ to further scale up TE device fabrication[11,12]. Currently, Mg$_3$(Sb,Bi)$_2$ is rendered as an ideal candidate for near-room-temperature applications, owing to the superb TE performance at low temperatures originating mainly from the intrinsically low $\kappa_L$ as a result of its complex crystal structure[13–16]. Meanwhile, the multivalley characteristics of n-type Mg$_3$(Sb,Bi)$_2$ can ensure outstanding electrical transport properties, beneficial to yielding high TE performance.

In recent years, the near-room-temperature performance of Mg$_3$(Sb,Bi)$_2$ has been improved by multiple strategies. By constructing the solid solutions with different Bi to Sb ratios, the band structure could be modified and the peak $zT$ value shifted towards a lower

[1]State Key Laboratory of New Ceramics and Fine Processing, School of Materials Science and Engineering, Tsinghua University, Beijing 100084, China. [2]Department of Materials Science and Engineering, Southern University of Science and Technology, Shenzhen 518055, China. [3]CAS Key Laboratory of Standardization and Measurement for Nanotechnology, CAS Center for Excellence in Nanoscience, National Centre for Nanoscience and Technology, Beijing 100190, China. ✉e-mail: hualu@mail.tsinghua.edu.cn; jingfeng@mail.tsinghua.edu.cn

temperature[17–19]. Moreover, the interest in interface engineering in $Mg_3(Sb,Bi)_2$ is growing, particularly focusing on the grain boundary engineering and two-phase interface engineering. On one hand, many strategies such as increasing sintering temperature[20], Mg-vapor annealing[15], and Cu[21] or Nb wetting phase[22] at grain boundaries, are employed to promote grain growth with the aim of optimizing the heterogeneity and electrical resistance of grain boundaries in $Mg_3(Sb,Bi)_2$[23,24]. However, this improvement is more pronounced at low temperatures probably due to the stronger carrier scattering at grain boundaries. On the other hand, by introducing highly conductive secondary phases, such as graphene nanoplatelets[25] and metal inclusions[26], at grain boundaries, the band structure at the grain boundaries can be bended, which helps to filter the low-energy electrons and enhance the electrical properties. Therefore, the introduction of highly conductive metal nanoparticles is expected to reduce the interfacial barrier in the $Mg_3(Sb,Bi)_2$ system.

Herein, high $zT$ values of ~ 2.0 were obtained in $Mg_3(Sb,Bi)_2$ by incorporating built-in metallic nano-inclusions. Given the good wettability at the grain boundaries of $Mg_3(Sb,Bi)_2$, Nb was added into the $Mg_3Sb_{1.5}Bi_{0.49}Te_{0.01}$ matrix prepared by mechanical alloying (MA) combined with spark plasma sintering (SPS). The heterogeneous interfaces induced by the nano-sized Nb inclusions together with the grain size controlled by the high-temperature sintering strategy played vital roles in modulating the electrical transport properties. It was inferred that the embedded Nb inclusions at grain boundaries could reduce the interfacial barriers, so the carriers can pass easily through the interfaces. Meanwhile, the $S$ can be obviously enhanced in the $Mg_3(Sb,Bi)_2$ with Ta addition, indicating that these built-in metallic nano-inclusions effectively modified the interfacial barriers and enhanced the contribution from high-energy electrons to transport properties. Along with the diminished $\kappa_L$ due to enhanced nanoparticle scattering, the $zT$ values of $0.1Nb/Mg_3Sb_{1.5}Bi_{0.49}Te_{0.01}$ significantly increased to 0.80 at 300 K and 2.04 at 798 K, respectively. A conversion efficiency of 15% under $\Delta T = 470$ K was attained for the single-leg device, showing great commercial prospects in a wide temperature range (300–798 K).

## Results

In general, both the carrier and phonon transports are significantly modulated by the Nb-rich secondary phases rather than Nb substitution. The electrical properties of the $xNb/Mg_3Sb_{1.5}Bi_{0.49}Te_{0.01}$ samples (Fig. 1) sintered at both the low and high temperatures show significant improvement after Nb addition, especially at 300–500 K. The near-room-temperature $\sigma$ is positively correlated with sintering temperature and Nb content (Fig. 1a). Notably, the $\sigma$ of $0.1Nb/Mg_3Sb_{1.5}Bi_{0.49}Te_{0.01}$ sintered at 1073 K is elevated to $7.6 \times 10^2$ S cm$^{-1}$ at room temperature, showing 30% enhancement compared to the unadded sample. The $\sigma$ of the $x = 0.2$ sample is even higher than that of the sample with $x = 0.1$. The absolute value of $S$ increases with temperature but remains almost constant after Nb addition (Fig. 1b).

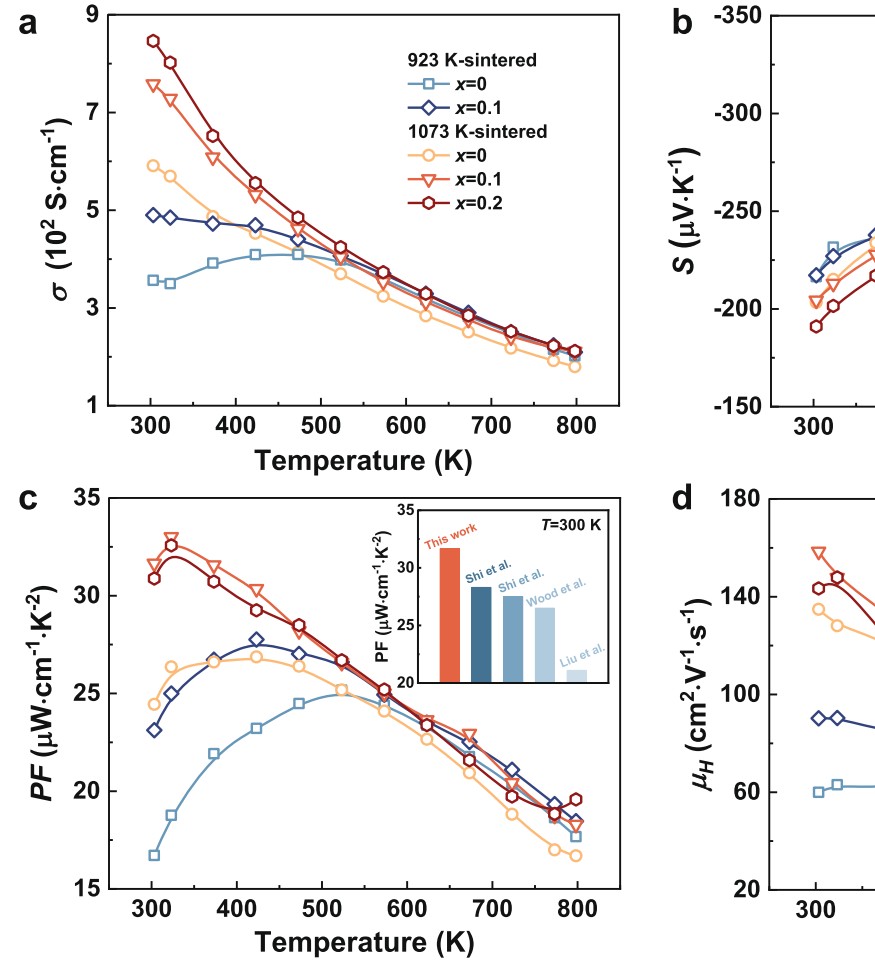

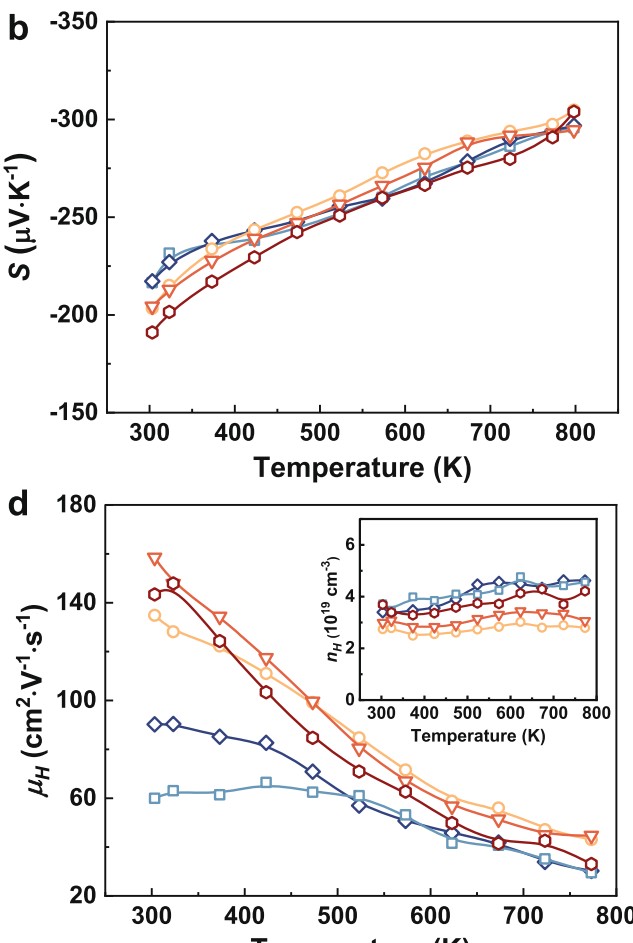

**Fig. 1 | The electrical transport properties of Nb-added samples. a** Temperature dependence of electrical conductivity. **b** Temperature dependence of Seebeck coefficient. **c** Temperature dependence of PF for $xNb/Mg_3Sb_{1.5}Bi_{0.49}Te_{0.01}$ ($x = 0$, 0.1 and 0.2) sintered at 923 K and 1073 K. The inset of **c** shows the comparison of PF in this work with previously reported results[15,21,27,28]. **d** Temperature dependence of Hall mobility for $xNb/Mg_3Sb_{1.5}Bi_{0.49}Te_{0.01}$ ($x = 0$, 0.1 and 0.2). The inset of **d** shows the temperature dependence of carrier concentration.

However, a slight decrease in $S$ is observed at low temperatures in the $x = 0.2$ sample. Due to the highly enhanced $\sigma$ with negligible changes to $S$, a high PF (over 31 $\mu$W cm$^{-1}$ K$^{-2}$) is obtained in 0.1Nb/Mg$_3$Sb$_{1.5}$Bi$_{0.49}$Te$_{0.01}$ at room temperature, as shown in Fig. 1c; this value ranks the highest among the data reported for Mg$_3$(Sb,Bi)$_2$[15,21,27,28] in earlier studies. Essentially, the $S$ contributes more to the high PF while the $\sigma$ is comparable to the values reported in other literatures, as demonstrated in Supplementary Fig. 1. The electrical properties of the sample with $x = 0.05$ and 0.15 are also shown in Supplementary Fig. 2. Clearly, with Nb addition, the temperatures corresponding to the highest $\sigma$ and PF also shift towards lower values, representing the optimization of electrical transport performance at low temperatures.

To provide clear insights into the superior electrical transport properties, the Hall carrier concentration ($n_H$) and mobility ($\mu_H$) were measured (Fig. 1d). The results of Hall measurement suggest that the $\mu_H$ of the $x = 0.1$ sample shows a remarkable increase at low temperatures, whereas this enhancement is obviously weakened above 450 K. Meanwhile, all the Nb-added samples sintered at 1073 K show enhanced $n_H$ within the entire temperature range compared to the unadded sample; the $n_H$ increases with increasing Nb content. Due to the increased $n_H$, the carrier scattering is enhanced, resulting in a decrease in $\mu_H$ for 0.2Nb/Mg$_3$Sb$_{1.5}$Bi$_{0.49}$Te$_{0.01}$. The weighted mobility ($\mu_W$) is also calculated, shown in Supplementary Fig. 3. The $x = 0.2$ sample maintains a high $\mu_W$ similar to that of $x = 0.1$, demonstrating noteworthy improvement compared to the unadded sample. On the contrary, the variation trend of $n_H$ with Nb content is different for the samples sintered at 923 K. In fact, the changes in $n_H$ and $\mu_H$ cannot be solely ascribed to the variable point defects (e.g. Mg vacancies[29] or doping atoms[27]) as thought conventionally, because the carrier scattering induced by point defects is usually more intense in the high-temperature range, which significantly affects the electrical properties at high temperatures. Mostly, the increased $n_H$ leads to a lower $S$. But there is an inconsistency between the trend of the $S$ and $n_H$ for the 1073 K-sintered samples. This may be attributed to the reduced interfacial barriers allowing more high-energy carriers to contribute to the $S$ (this will be anatomized later). Therefore, a higher $S$ is still observed in $x = 0.1$ sample with an increased $n_H$. In general, according to $\sigma = n_H e\mu_H$, where $e$ is the electronic charge, the increased $n_H$ and $\mu_H$ contribute in tandem to enhancing $\sigma$. As a result, Nb addition not only provides extra electrons but also enhances $\mu_H$ near room temperature. By contrast, the reduced mobility at high temperatures is probably assigned to the enhanced electron scattering. Meanwhile, the slight decrease in $S$ mainly stems from the increased $n_H$. Similarly, a more pronounced decrease in $S$ is observed due to higher $n_H$ in the $x = 0.2$ sample. Nevertheless, the augmented electron concentration donated by Nb still guarantees higher $\sigma$ (a 17% increase at 798 K), leading to enhanced PF at high temperatures. This might be ascribed to a small amount of Sb atoms being consumed from the matrix caused by the formation of Nb$_3$Sb phase.

The enhanced charge carrier transport is closely associated with microstructures. As shown in Fig. 2a, the grain size of the 923 K-sintered samples is estimated to be about 2 μm. The grain growth seems insensitive to Nb addition (Fig. 2a, b), but the $\sigma$ at room temperature is improved. For the 1073 K-sintered samples (Fig. 2c, d), the grain size increases more than threefold in value (~7.2 μm), compared to that of the 923 K-sintered samples. Interestingly, by elevating sintering temperature, only a slighter increase in grain size (from 2.3 to 3.7 μm) was observed for the $x = 0.1$ sample. In spite of smaller grain size, the $\mu_H$ of the Nb-added samples is still retained a high level (158.4 cm$^2$ V$^{-1}$ s$^{-1}$), even higher than that of the unadded samples (134.8 cm$^2$ V$^{-1}$ s$^{-1}$). This result conflicts with the conventional view that the room-temperature $\sigma$ increases with grain size, because of the reduced density of electrically resistive grain boundaries[15,20,24]. The smaller grain size in the Nb-added samples sintered at 1073 K contributes to enhancing the grain boundary resistance, but the presence of Nb

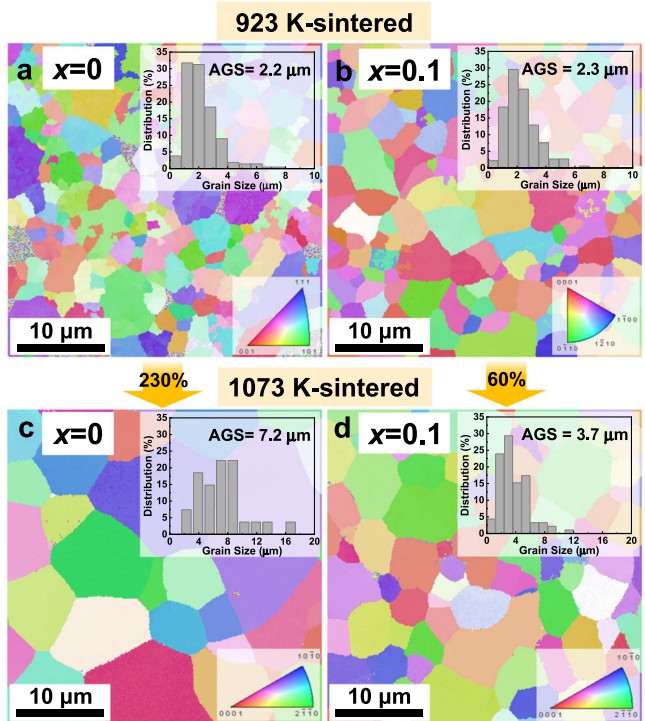

**Fig. 2 | Grain size evolution induced by Nb inclusions.** Electron backscatter diffraction (EBSD) crystal-orientation maps of $x$Nb/Mg$_3$Sb$_{1.5}$Bi$_{0.49}$Te$_{0.01}$ ($x = 0$ and 0.01) sintered at 923 K (**a**, **b**) and 1073 K (**c**, **d**). The insets are the corresponding statistics on the grain size distribution.

inclusion is able to cancel the negative effect of grain boundaries on electrical properties near room temperature. As shown in the X-ray diffraction (XRD) patterns (Supplementary Fig. 4), extra Bragg reflection peaks are detected at 38.4° in the Nb-added samples. The energy-dispersive X-ray spectroscopy (EDS) elemental mapping on the fracture surface (Supplementary Fig. 5) also confirms the existence of Nb-rich secondary phases in the Nb-added samples. The Nb inclusion is further investigated by a microscope equipped with an EDS detector in the $C_s$-corrected high-angle annular dark-field scanning transmission electron microscopy (HAADF-STEM) mode (Fig. 3a–e). It is found that the Nb-rich secondary phases distribute randomly at grain boundaries. The atomic ratios of the central region of the Nb inclusion are shown in Fig. 3f, matching well with the chemical composition of Mg$_{4.5}$Nb$_{86.0}$Sb$_{9.5}$. Furthermore, the EDS analysis also indicates that the Nb atoms do not enter the lattice of the matrix, as shown in Supplementary Fig. 6. More Nb-rich inclusions, about ten to hundreds of nanometers in diameter, are observed in Supplementary Fig. 7. These inclusions help to pin grain boundaries, hindering grain boundary migration and suppressing grain growth.

Back to the question: how does Nb benefit the electrical transport? Considering the facts that increased grain boundaries enhance the interface resistance and Nb inclusions increase the number of interfaces, it is not difficult to infer that the interfaces between Nb and matrix should own lower resistance than the grain boundaries. Therefore, the embedded Nb inclusions at grain boundaries should help to reduce the interfacial barriers and weaken the grain boundary scattering, which promotes carrier transport, as schematically shown in Fig. 3g. Essentially, it may be derived from the weak interface scattering due to the good wettability of Nb to the grains[22]. It is known that interface scattering is more effective in the low-temperature range. This also explains why the improvement in $\mu_H$ becomes negligible in the high-temperature range. In addition, the resulting metal-semiconductor interfaces modulate the scattering mechanisms. In

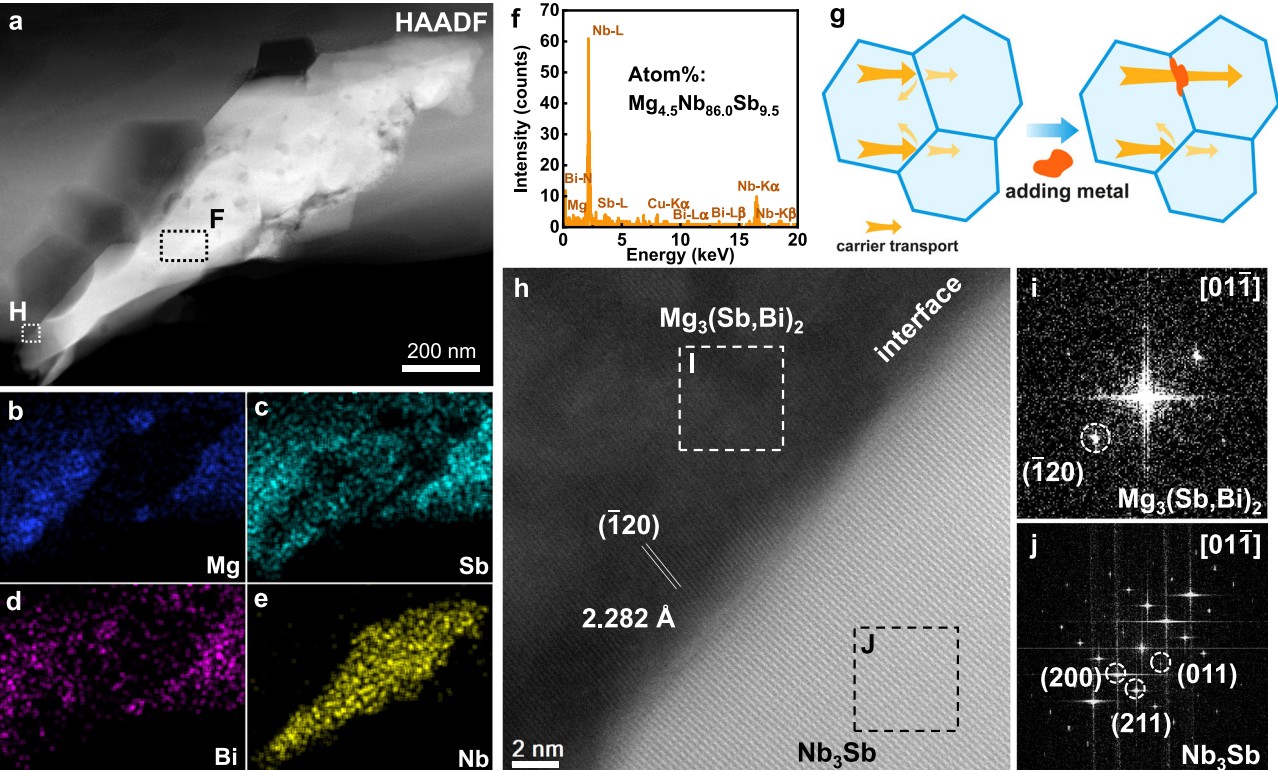

**Fig. 3 | Microstructure and interface analysis. a** HAADF-STEM mapping of 0.1Nb/ $Mg_3Sb_{1.5}Bi_{0.49}Te_{0.01}$ sintered at 1073 K, and corresponding EDS maps for (**b**) Mg, (**c**) Sb, (**d**) Bi and (**e**) Nb. **f** EDS analysis for a Nb-rich secondary phase inclusion from a random region F in **a**. **g** Schematic illustration of different carrier transport behaviors at grain boundaries and metallic inclusions. **h** An HRTEM image of the interface between the matrix and Nb-rich secondary phase from the region H in **a** with the (**i**) FFT image of the boxed region I and (**j**) FFT image of region J in **h**.

0.1Nb/$Mg_3Sb_{1.5}Bi_{0.49}Te_{0.01}$, the absolute value of the temperature exponent derived from the Hall mobility is higher than that of the matrix. It indicates that the grain boundary scattering mechanism is partially involved for the large-grain samples. The modified interfacial effect by metallic inclusions cancels the decrease in mobility caused by grain boundary scattering, making the acoustic phonon scattering dominant. Figure 3h presents the high-resolution transmission electron microscopy (HRTEM) image of the interface between the matrix and Nb inclusions. The corresponding fast Fourier transform (FFT) patterns (Fig. 3i, j) illustrate the crystal structures of $Mg_3(Sb,Bi)_2$ and $Nb_3Sb$ viewed in the [01$\bar{1}$] zone axis. The Nb-Sb binary phase diagram can serve as evidence for the presence of stable $Nb_3Sb$ phase (Supplementary Fig. 8a)[30]. However, no peaks of the $Nb_3Sb$ phase are observed in the XRD patterns (Supplementary Fig. 4), indicating that only a small amount of $Nb_3Sb$ is formed at the interface. The Sb in the matrix may diffuse to the Nb inclusion, leading to the formation of new secondary phases near the interfaces. The $Nb_3Sb$ phase has a resistivity of $7 \times 10^{-7}\,\Omega\,m$ at 298 K[31], which is slightly higher than the value of $1.6 \times 10^{-7}\,\Omega\,m$ for Nb metal[32] but much lower than that of the matrix. These composition-transition interfaces with high conductivity further support the modulation of interfacial barriers by Nb inclusions. Similar interfacial structures between the matrix and the metallic inclusions are displayed in Supplementary Fig. 9. The inclusions perhaps have a core-shell-like structure with Nb as the main body and a small amount of $Nb_3Sb$ as the coating layer, as demonstrated in Supplementary Fig. 8b. As there is no orientation relationship between the $Nb_3Sb$ phase and the matrix, the Nb-rich inclusions are unlikely to be in situ informed. The modulation of the scattering mechanisms by Nb inclusions remarkably improves the mobility. Besides, due to the formation of the $Nb_3Sb$ phase near the interface, a small amount of Sb atoms in the matrix would be inevitably consumed by Nb. As shown in

Supplementary Fig. 10, the results of electron probe microanalysis (EPMA) confirmed that the atomic ratio of Sb/Mg in the matrix of $x = 0.1$ sample is lower than that of the unadded sample. The decreased Sb/Mg ratio may suppress the formation of Mg vacancies, which is beneficial to increasing the $n_H$ and hence the $\sigma$ of $Mg_3(Sb,Bi)_2$. Moreover, the investigation of the Nb/$Mg_3SbBi$ interface also confirms that Nb is in good contact with the matrix without macroscopic secondary phases[33,34].

The Nb nano-inclusions not only influence the carrier transport but also interact with phonons. The temperature dependence of total and lattice thermal conductivity is shown in Fig. 4a and Supplementary Fig. 2. The $\kappa$ and $\kappa_L$ decrease first and then increase upon the addition of Nb. A relatively low $\kappa$ of $0.45\,W\,m^{-1}\,K^{-1}$ is obtained for 0.1Nb/ $Mg_3Sb_{1.5}Bi_{0.49}Te_{0.01}$ sintered at 1073 K. On one hand, the $\kappa$ and $\kappa_L$ are significantly reduced due to the abundant vacancies and vacancy clusters as a consequence of the loss of volatile elements at high sintering temperatures (detailed discussion is provided in our previous work)[35]. On the other hand, the $\kappa_L$ is further reduced by nanoparticle scattering. The sizes of Nb secondary phase are averaged about 200 nm in diameter but are distributed from 10 to 1000 nm, which cover a large range of phonon mean free paths. The variable $\kappa_L$ is probably related to the different interfacial thermal resistance originating from the layered or granular bonding between metal and matrix. Notably, it was reported that the $\kappa$ increased when the thin wetting Nb layer was formed at grain boundaries[22]. In this case, the two-phase model equivalent to a one-dimensional series circuit can be utilized to understand the increase in $\kappa$[24], which probably stems from the reduction in interfacial thermal resistance. When Nb exists in the form of secondary phases (from nano to sub-micron scales), the additional scattering centers induced by these nano-inclusions contribute to the reduction in $\kappa_L$ by suppressing the propagation of

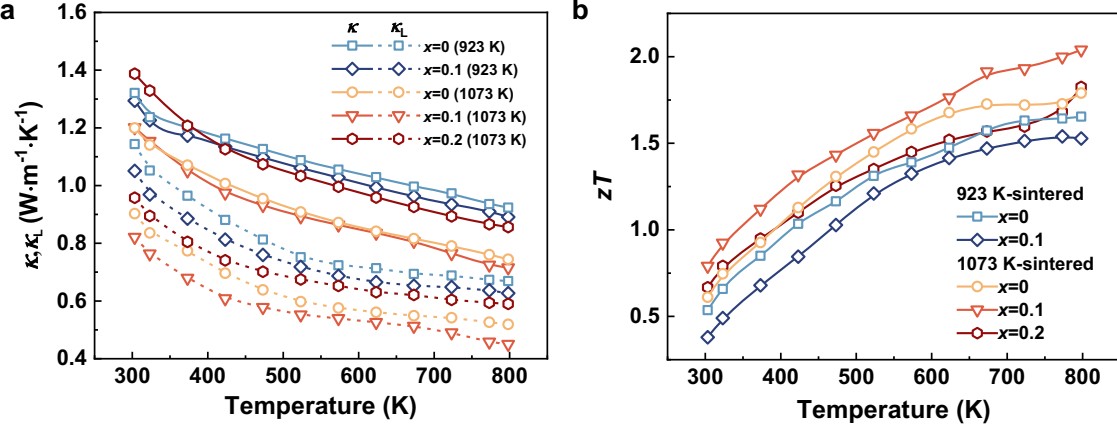

**Fig. 4 | Thermal conductivity and figure-of-merit $zT$ of Nb-added samples.** Temperature dependence of (**a**) total thermal conductivity and lattice thermal conductivity, and (**b**) $zT$ values for $x$Nb/Mg$_3$Sb$_{1.5}$Bi$_{0.49}$Te$_{0.01}$ sintered at 923 K and 1073 K.

phonons. However, the $\kappa$ significantly increases when the amount of Nb increases to more than 0.1; higher fractions of thermally conducting Nb phases are more likely to affect the $\kappa$ through its intrinsic nature. When the contribution from the high $\kappa$ of Nb phase (53.7 W m$^{-1}$ K$^{-1}$) overweighs the reduction in $\kappa$ by phonon scattering, the $\kappa$ and $\kappa_L$ will rise. Moreover, due to the more pronounced Mg volatilization during high-temperature sintering, no Mg-rich phase is observed in the unadded and Nb-added samples, which is confirmed by the EDS analysis (Supplementary Figs. 5 and 11), excluding the effect of excessive Mg on $\kappa$. The total TE performance is enhanced by Nb nano-inclusions in conjunction with high-temperature sintering, as shown in Fig. 4b. The room-temperature $zT$ value increases from 0.38 to 0.80 and the peak $zT$ value is elevated from 1.53 to 2.04 for 0.1Nb/Mg$_3$Sb$_{1.5}$Bi$_{0.49}$Te$_{0.01}$ due to the increased PF and reduced $\kappa$, showing ~110% and 33% enhancement, respectively. Moreover, the TE properties of the as-prepared samples exhibit good repeatability, which can be confirmed by the data in Supplementary Fig. 12.

Furthermore, since Nb and tantalum (Ta) are mainly found in tantalite and are symbiotic with each other, the Ta is expected to exhibit similar behaviors. As expected, similar enhancement in the TE performance was discovered in the Ta-added Mg$_3$(Sb,Bi)$_2$ sintered at 1073 K. Figure 5a depicts the temperature-dependent electrical conductivity and Seebeck coefficient of 0.1Nb/Mg$_3$Sb$_{1.5}$Bi$_{0.49}$Te$_{0.01}$ and 0.1Ta/Mg$_3$Sb$_{1.5}$Bi$_{0.49}$Te$_{0.01}$ prepared under the same conditions. Different from the significant increase after Nb addition, the $\sigma$ only increases slightly after Ta addition. Yet it is worth noting that the $S$ increases with Ta addition, especially at high temperatures. Based on the simultaneous optimization of $\sigma$ and $S$, the PF of 0.1Ta/Mg$_3$Sb$_{1.5}$Bi$_{0.49}$Te$_{0.01}$ is up to 32 $\mu$W cm$^{-1}$ K$^{-2}$ at 323 K and close to 20 $\mu$W cm$^{-1}$ K$^{-2}$ at 798 K, as shown in Fig. 5b. Interestingly, the improvement in electrical properties near room temperature is the same as the case relevant to Nb but less dramatic, while the effect of Ta becomes more pronounced at high temperatures. As shown in Supplementary Table 1, both $n_H$ and $\mu_H$ of the Ta-added sample slightly increase at room temperature, but are less than the values of the Nb-added sample. Similar to the Nb-added samples, nanoscale Ta metal inclusions randomly distribute at grain boundaries in 0.1Ta/Mg$_3$Sb$_{1.5}$Bi$_{0.49}$Te$_{0.01}$, as demonstrated in Supplementary Figs. 13–15. The reason may be that both Nb and Ta are refractory and have good chemical stability. The scanning electron microscope (SEM) images of the fracture surfaces shown in Supplementary Fig. 16 display the grain size of Nb-added and Ta-added samples. The grain size of the Ta-added sample approximately decreases to the level of that of the Nb-added samples, which is also attributed to the pinning effect of inclusions at grain boundaries. The embedded Ta nano-inclusions at grain

boundaries should have the similar effect as Nb. However, no Ta-rich secondary phases similar to Nb$_3$Sb were formed under the present experimental conditions; in fact, that no Ta-Sb binary phase diagram was reported might account for why the Ta addition did not affect the carrier concentration as in the case of the Nb-added samples.

Benefiting from the marginally increased $n$, the contribution to the $S$ from the inclusions becomes easier to distinguish. The built-in Nb/Ta nano-inclusions may scatter the low-energy carriers more efficiently via modifying the interfacial barriers[36,37]. As mentioned above, the secondary phases near the interface would reduce interfacial potential barriers. It is speculated that these barriers of metal-semiconductor interfaces are lower than that of Mg-deficient grain boundaries, as demonstrated in Fig. 5c. The higher barriers at grain boundaries ($E_{b1}$) in the unadded samples would block low-energy and part of high-energy carriers. By incorporating Ta inclusions, the interfacial barrier is reduced from $E_{b1}$ to $E_{b2}$, allowing more high-energy carriers to pass through. The contribution from high-energy carriers to transport properties is enhanced, giving rise to a higher $S$. In the case for Nb-added samples, this modified interfacial barrier turns out to optimize $\sigma$ without sacrificing $S$, thereby leading to enhanced PF. The $S$ may be a synergistic result of the above effect and increased $n_H$. The Pisarenko plots shown in Supplementary Fig. 17 exhibit the contribution from the modified interfacial barrier to $S$, where the data points deviate upward slightly. Recently, it has been reported that the Mo addition also increases the $\sigma$ of Mg$_3$(Sb,Bi)$_2$-based materials[38], indicating the great potential of other transition metals. However, the enhanced $\sigma$ by Mo addition is mainly due to the grain boundary effect, different from those analyzed above for the Nb or Ta inclusions.

The incorporation of Nb and Ta nano-inclusions into the Mg$_3$(Sb,Bi)$_2$ matrix helps to modify the interfacial barriers, producing similar favorable modulation results. As a consequence of the optimized electrical performance, a high $zT$ value of 2.06 is obtained in 0.1Ta/Mg$_3$Sb$_{1.5}$Bi$_{0.49}$Te$_{0.01}$ at 798 K, as shown in Fig. 6a. However, due to the higher $\kappa$ (57.5 W m$^{-1}$ K$^{-1}$) as well as weaker phonon scattering originating from the larger size of Ta inclusions, the effect of Ta inclusions on reducing $\kappa$ is limited (Supplementary Fig. 18). The $zT$ values of Ta-added samples within the low- and middle-temperature range are slightly lower than that of Nb-added samples. After adding these metallic inclusions into the matrix, the $zT$ values increase over the entire temperature range, thereby boosting the average $zT$ value ($zT_{ave}$) calculated using the following formula:

$$zT_{ave} = \frac{\int_{T_c}^{T_h} zT\,dT}{T_h - T_c} \tag{1}$$

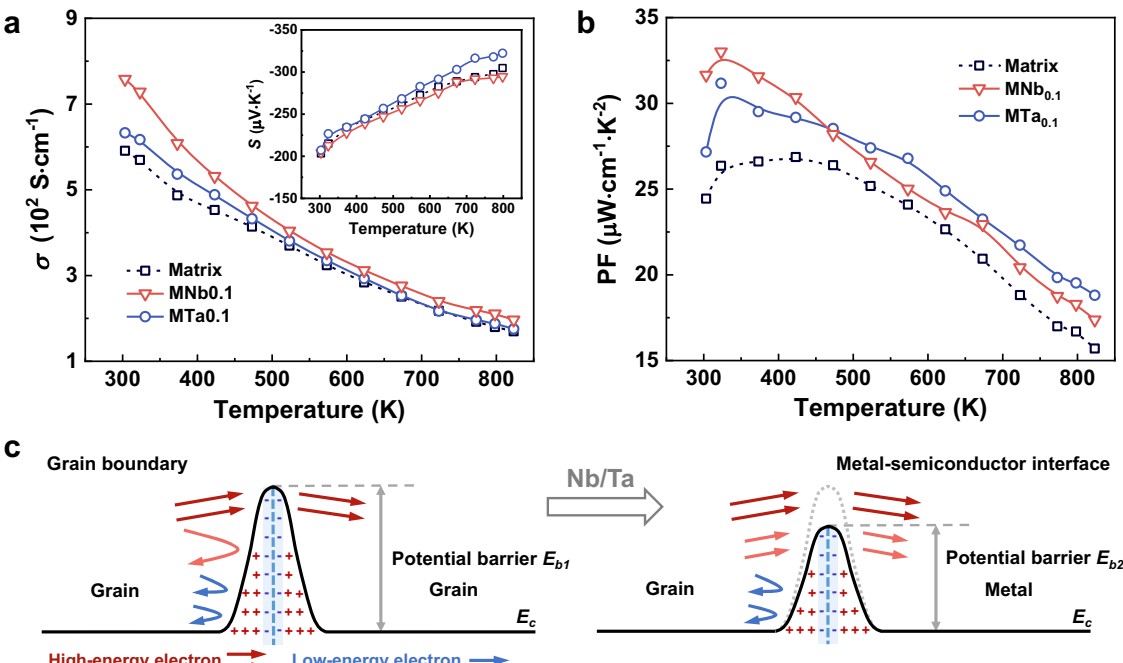

**Fig. 5 | The electrical transport properties of Ta-added samples and illustration of the modified interfacial barrier.** Temperature dependence of (**a**) electrical conductivity and Seebeck coefficient and (**b**) PF for $0.1Nb/Mg_3Sb_{1.5}Bi_{0.49}Te_{0.01}$ and $0.1Ta/Mg_3Sb_{1.5}Bi_{0.49}Te_{0.01}$. **c** Schematic illustration of interfacial barriers near the grain boundary region and metal-semiconductor interface region; $E_b$ is the interfacial potential barrier and $E_c$ is the energy of the conduction band minimum.

where $T_h$ and $T_c$ are the temperatures of the hot side and cold side, respectively. The $zT_{ave}$ values reach 1.57 and 1.51 within the temperature range of 300–798 K for $0.1Nb/Mg_3Sb_{1.5}Bi_{0.49}Te_{0.01}$ and $0.1Ta/Mg_3Sb_{1.5}Bi_{0.49}Te_{0.01}$, respectively. The $zT_{ave}$ values of the Nb-added sample in the temperature range 300–773 K and 300–573 K are also calculated, which are 1.54 and 1.30, respectively. Compared to the previously reported results for $Mg_3(Sb,Bi)_2$, the $zT_{ave}$ values are at the state-of-the-art level and even superior to $Bi_2(Te,Se)_3$, which is competitive near room temperature (Fig. 6b)[20,27,39–43]. Based on such a high TE performance for the optimized sample, i.e. $0.1Nb/Mg_3Sb_{1.5}Bi_{0.49}Te_{0.01}$ sintered at 1073 K, a single-leg device was fabricated with the $Fe_7Mg_2Cr$ interface and sintered copper electrode to evaluate the TE energy conversion efficiency. As shown in Fig. 6c and Supplementary Fig. 19, a high efficiency of around 15% is obtained for a single leg under a $\Delta T$ of 470 K. The excellent reproducibility of the efficiency measured by the single legs is shown in Supplementary Figs. 20 and 21. The theoretical efficiency of the single-leg device reaches 18% as shown in Supplementary Fig. 22, but both the measured output power and heat flow deviate from the predicted value, indicating additional efforts are needed to optimize the contact layer and control the thermal radiation. In addition, a continuous measurement on the output properties (the inset of Fig. 6d and Supplementary Fig. 23) reveals that the single leg can operate over 120 h at a $\Delta T = 470$ K with an efficiency higher than 14%, showing good service stability. Figure 6d exhibits the high-efficiency values reported in $Mg_3(Sb,Bi)_2$ system in recent years, indicating the efficiency achieved in this work rivals other multiple-leg modules but has a wider application temperature range. It is worth noting that, due to the promoted electrical transport by the metallic nano-inclusions, a competitive value is obtained in the low-temperature range, surpassing that of the classical bismuth telluride with broad prospects for commercial applications in electronic cooling.

## Discussion

In summary, a high peak $zT$ value of 2.04 at a relatively low temperature (798 K) and a remarkable $zT_{ave}$ (300–798 K) of 1.57 for $Mg_3(Sb,Bi)_2$ were reported in this work. The fabricated single-leg device showed a high TE conversion efficiency of ~15%, which rivaled those of well-designed modules. The significantly improved TE performance was attributed to the modulation of charge carriers and phonons transport via introducing Nb/Ta inclusions at grain boundaries. The reduced interfacial barriers significantly increased the electrical conductivity in the low-temperature range. Besides, the Nb and Ta inclusions not only contributed to increasing $n_H$ but also led to remarkably enhanced $\mu_H$ near room temperature. As a result, the PF was significantly increased, especially at low temperatures. Combined with the effective phonon scattering effect caused by a small number of metallic inclusions, the lattice thermal conductivity was also obviously diminished. Our findings provide a new strategy to deal with the high grain boundary resistance in $Mg_3(Sb,Bi)_2$-based materials, particularly demonstrating great potential in near-room-temperature applications.

## Methods
### Synthesis

Magnesium turnings (99.9%), antimony shots (99.999%), bismuth shots (99.999%), tellurium powder (99.999%), niobium powder (99.9%, 325 mesh) and tantalum powder (99.9%, 325 mesh) were weighed in accordance with the nominal stoichiometry of $Mg_{3.2}Sb_{1.5}Bi_{0.49}Te_{0.01}$ and $Mg_{3.1}X_{0.1}Sb_{1.5}Bi_{0.49}Te_{0.01}$ ($X$ = Nb and Ta) and then ball-milled for 8 h in the SPEX 8000D Mixer/Mill ball mill (SPEX, Metuchen, NJ, United States). To explore the effect of Nb on electrical transport properties, the powder with the composition of $Mg_{3.1}Nb_xSb_{1.5}Bi_{0.49}Te_{0.01}$ ($x$ = 0.05, 0.15, and 0.2) was also prepared. The ball-milled powder was sintered by spark plasma sintering using the SPS 211Lx Spark Plasma Sintering Machine (Fuji Electronic Industrial Co., Tsurugashima, Saitama, Japan) at 923 K and 1073 K for 20 min under an axial pressure of 50 MPa. To fabricate the single-leg device, the sintered bulk, TE interface material (TEiM) powders and Cu powders (99.9%) were sandwiched in a graphite die and then sintered at 873 K for 10 min under a pressure of 50 MPa. The TEiM powders were generated by ball milling after mixing Fe powder (99.9%), Mg turnings

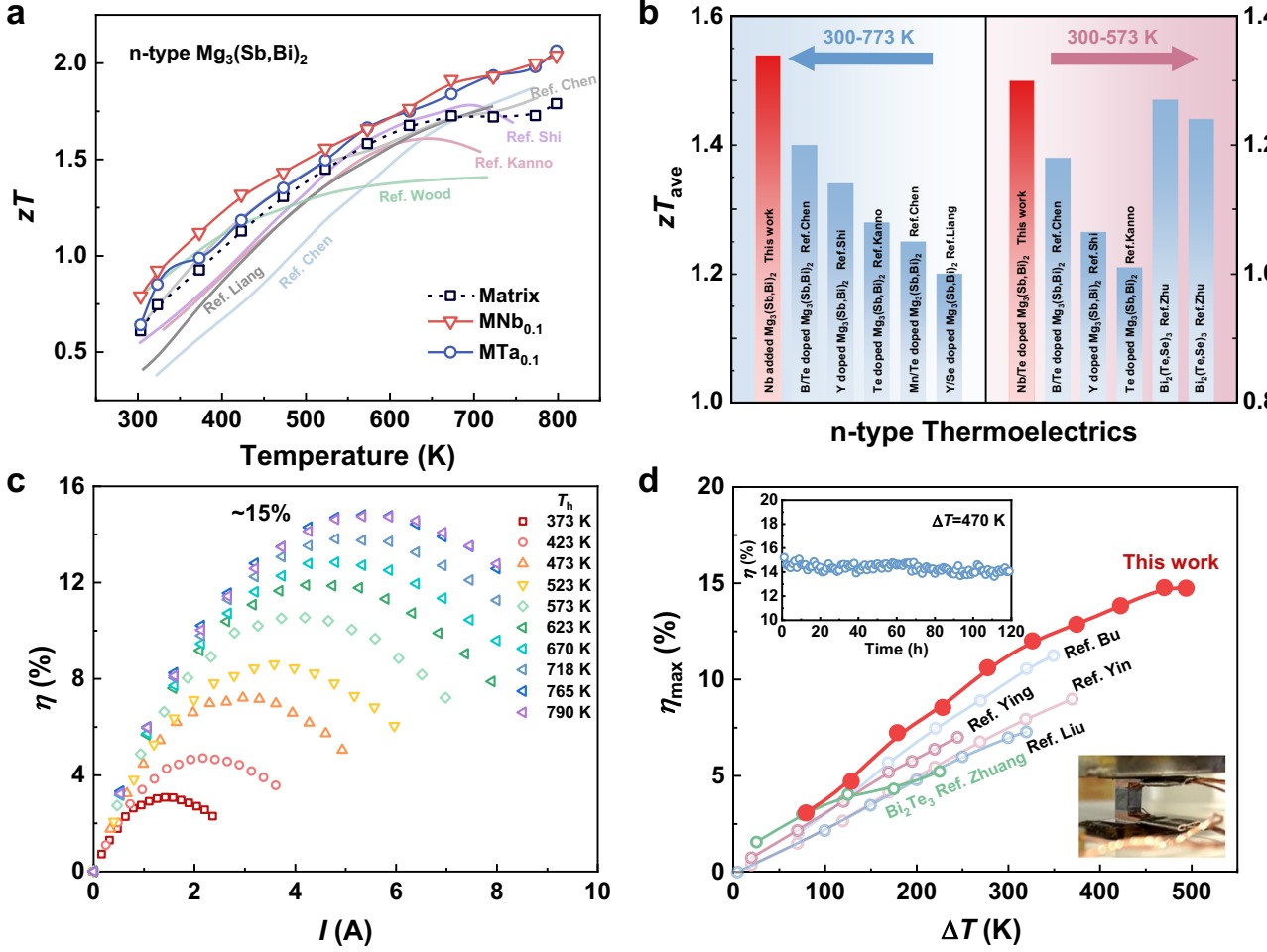

**Fig. 6 | Comparison of figure-of-merit *zT* and conversion efficiency with previous studies. a** Temperature dependence of *zT* values for 0.1Nb/ $Mg_3Sb_{1.5}Bi_{0.49}Te_{0.01}$ and $0.1Ta/Mg_3Sb_{1.5}Bi_{0.49}Te_{0.01}$ in comparison with the state-of-the-art n-type $Mg_3(Sb,Bi)_2$[15,20,27,39–41]. **b** Comparison of average *zT* in the temperature range of 300–773 K and 300–573 K for $0.1Nb/Mg_3Sb_{1.5}Bi_{0.49}Te_{0.01}$ and other n-type thermoelectrics[20,27,39–43]. **c** The measured conversion efficiency of the single-leg device in the temperature range of 373–790 K and the cold side temperature is 295 K. **d** Comparison of the maximum conversion efficiency among the single-leg device in this work, other $Mg_3(Sb,Bi)_2$ modules, and $Bi_2Te_3$ module under a series of temperature difference ($\Delta T$)[21,45–48]. The inset in **d** is the optical image of the fabricated TE single-leg device in this work and the time-dependent efficiency for the single-leg device under $\Delta T = 470$ K.

(99.9%), and Cr powder (99.5%). The thicknesses of the sintered bulk and TEiM layers were 8 and 1.5 mm, respectively.

## Characterization

The phase purity of the material was determined by powder X-ray diffraction (XRD) using the Bruker D8 Advance X-ray diffractometer (Bruker AXS GmbH, Karlsruhe, Germany) with Cu-Kα radiation. The grain size and crystal orientation were determined via electron backscatter diffraction (EBSD) by the ULVAC-PHI 710 Scanning Auger spectrometer (ULVAC-PHI, Inc., Chigasaki, Japan). Scanning transmission electron microscopy (STEM) observations were performed on the Spectra 300 double-$C_s$-corrected transmission electron microscope (Thermo Fisher Scientific Inc.) equipped with a field-emission electron source, operating at an accelerating voltage of 300 kV. Atomic resolution energy-dispersive X-ray spectroscopy (EDS) elemental mapping and analysis were conducted using the Super-X EDS detector. The morphology and microstructure were characterized by field-emission scanning electron microscopy (FESEM, Zeiss Merlin, Germany). The fracture surface morphology and elemental distribution were characterized by field-emission scanning electron microscope (FE-SEM, Gemini 2, Zeiss, Germany) equipped with an EDS detector. The quantitative analysis of

elements was studied via electronic probe microscopic analysis (JXA-8230, JEOL, Japan).

## Thermoelectric property measurements

The Seebeck coefficient ($S$) and conductivity ($\sigma$) were determined using a ZEM-3 Seebeck coefficient/electric resistance measuring system (ULVAC-RIKO Inc., Yokohama, Japan). Thermal conductivity ($\kappa$) was calculated using the equation $\kappa = DC_p\rho$, where the thermal diffusivity ($D$) was measured with the LFA 457 laser flash apparatus (Netzsch GmbH, Selb, Germany). The specific heat capacity ($C_p$) was derived using the Dulong–Petit law, and the density ($\rho$) was measured in accordance with the Archimedes method. The electronic thermal conductivity ($\kappa_e$) was calculated based on the Wiedeman–Franz law ($\kappa_e = \sigma LT$), where the Lorenz factor ($L$) was estimated using the equation $L = 1.5 + \exp(-\frac{|S|}{116})$[44]. The Hall coefficient ($R_H$) was determined using the Hall measurement system (ResiTest 8340DC, Tokyo, Japan). The Hall carrier concentration ($n_H$) and mobility ($\mu_H$) were calculated using the equations $n_H = 1/(eR_H)$ and $\mu_H = \sigma R_H$, respectively. The conversion efficiency ($\eta$) of the single-leg device was calculated in accordance with the equation $\eta = P/(P + Q_c) \times 100\%$, where the output power ($P$) and cold side heat flow ($Q_c$) were simultaneously measured with the Mini-PEM testing

system (Ulvac-Riko, Japan). The theoretical conversion efficiency was simulated using the COMSOL Multiphysics software.

## Data availability

The data that support the findings of this study are available from the corresponding author on request.

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

## Acknowledgements

This work was supported by The Basic Science Center Project of National Natural Science Foundation of China (grant no. 52388201) and the National Key R&D Program of China (no. 2023YFB3809400).

## Author contributions

J.-F.L., J.-W.L. and H.-L.Z. conceptualized this work. J.-W.L. performed the synthesis and thermoelectric property measurements. H.H. performed the EBSD measurement. Q.Z. carried out the STEM measurement. Z.H. and W.L. contributed to the preparation and measurement of single-leg devices. J.-W.L., B.S. and L.C. performed the structure characterizations. J.-W.L., H.-L.Z., J.Y., H.L. and Y.J. helped the analysis of the thermoelectric property data. J.-W.L., H.-L.Z., J.Y. and J.-F.L. wrote the manuscript. All authors participated in the data analysis and the manuscript editing.

## Competing interests

The authors declare no competing interests.
