## [Peer Review File · Nature Communications]

Wide-temperature-range thermoelectric n-type $\text{Mg}_3(\text{Sb},\text{Bi})_2$ with high average and peak zT valuesEditorial Note: Parts of this Peer Review File have been redacted as indicated to remove third-party material where no permission to publish could be obtained.

REVIEWER COMMENTS

Reviewer #1 (Remarks to the Author):

The paper presents a comprehensive investigation into the impact of metallic inclusions (specifically, Nb and Ta) on the thermoelectric properties of n-type $\text{Mg}_3(\text{Sb,Bi})_2$ -based material. The authors identified the presence of a Nb-rich second phase at grain boundaries, which leads to grain growth inhibition and a substantial enhancement of electric transport properties, attributed to reduced interface potential barriers. Impressively, this study achieves a zT of 0.8 at room temperature and a notable peak zT value of 2.0 at 798K. Moreover, a single leg device demonstrates a remarkable energy conversion efficiency of 14% across a temperature differential of 470K. The research results are robust, and the manuscript is well-organized. However, certain aspects and major concerns have to be addressed prior to consideration for publication in Nature communications.

1. Wang et al. published a similar study on the performance optimisation of $\text{Mg}_3(\text{Sb,Bi})_2$ thermoelectrics through Mo addition using the same ball-milling and with SPS method (Adv. Engng Mater. 2023, 2301667). Their results also demonstrated the increased electron transport properties and decreased lattice thermal conductivities in Mo-added samples. Meantime, many studies have focused on the doping of transition metal (alloys) in the $\text{Mg}_3(\text{Sb,Bi})_2$ system. The authors need to conduct further literature review and clarify the novelty of this work.
2. This work mainly studied $\text{Mg}_3\text{Sb}_{1.5}\text{Bi}_{0.49}\text{Te}_{0.01}$ based samples. However, $x\text{Nb}/\text{Mg}_3(\text{Sb,Bi})_2$ and similar expressions were used throughout the manuscript, which may mislead readers and to some extent was over claiming.
3. Figure 1b and the inset of Figure 1d present an inconsistency between the trend of the Seebeck coefficient and carrier concentration. Given their leading roles in thermoelectric behavior, it is essential to address this inconsistency comprehensively. The introduction of a Pisarenko line encompassing all compositions, along with an explanatory analysis is required.
4. Lines 97-99: "Clearly,...optimization of electrical transport performance at low temperatures". Only two compositions of Nb-added samples, i.e. 0.1 Nb and 0.2 Nb, were prepared in this study, thus, the statement here was not solidly supported. Generally, this study delivered only three compositions of Nb/Ta-added samples, which was not as convincing as one may expected from a comprehensive study published on Journals such as Nature Communications.
5. Discussion in the paragraph from line 100 was not very rigorous and lacks supports from literature studies. E.g. "In fact, the changes in nH and uH cannot be ascribed to the variable point defects (e.g. Mg vacancies) as thought conventionally." Refs should be cited and explanations should be included. "This might be ascribed to the Sb deficiency in the matrix caused by the formation of the Nb_3Sb phase." Links to related results should be made and no clear evidence on Sb deficiency in the matrix was available in the current manuscript and SI.
6. Lines 208-210: similar to above, how does the Sb deficiency affect nH and σ ? Are there any reported studies? Explanations should be made in this part.
7. Since a substantial portion of a Nb-rich second phase is found at grain boundaries, as confirmed by TEM, SEM, and XRD analyses, it is imperative to provide the thermal transport

properties of this Nb-rich compound.

8. More experiments and more compositions are suggested to be performed to evaluate k and k_L . Why the k and k_L values contrasted remarkably between $0.1\text{Nb}/\text{Mg}_3\text{Sb}_{1.5}\text{Bi}_{0.49}\text{Te}_{0.01}$ and $0.2\text{Nb}/\text{Mg}_3\text{Sb}_{1.5}\text{Bi}_{0.49}\text{Te}_{0.01}$ but negligibly between $0.1\text{Nb}/\text{Mg}_3\text{Sb}_{1.5}\text{Bi}_{0.49}\text{Te}_{0.01}$ and $\text{Mg}_3\text{Sb}_{1.5}\text{Bi}_{0.49}\text{Te}_{0.01}$? What are the optimised compositions?

9. Are there any links between the energy-filtering effect in paragraph (lines 278-290) and previous discussion & microstructures?

10. In conclusions: "Combined with the effective phonon scattering effect caused by the metal inclusions, the lattice thermal conductivity was also obviously diminished." There seems no rigorous evidence on the effective phonon scattering effect caused by the metal inclusions (e.g. Figure 4a). The 0.1Nb and 0.2Nb samples depicted different trends on scattering phonons.

11. A central innovation of this work revolves around the reduction of potential barriers between the grains due to the metallic inclusions. To bolster the clarity and significance of the findings, a fabrication of Nb contact layer on the optimized $\text{Mg}_3(\text{Sb},\text{Bi})_2$ materials with an in-depth investigation of the transport properties of the $\text{Nb}/\text{Mg}_3(\text{Sb},\text{Bi})_2$ interface is recommended.

12. As the author state that Ta was expected to exhibit similar behaviors. In the inset of Figure 5a, the high temperature Seebeck coefficient of $\text{Ta}_{0.1}$ -added sample was improved significantly while the addition of Nb didn't show a similar effect. And also, in Figure S13, the addition of Nb and Ta have opposite effects on the lattice thermal conductivity. The author should clarify and have in depth discussion on it.

13. It is suggested to predict the performance of the single leg device based on the thermoelectric properties of the optimized materials. A comparison between predicted and measured results would not only elucidate the quality of device fabrication but also provide insights into potential measurement errors.

14. As for the thermoelectric performance of $0.1\text{Nb}/\text{Mg}_3(\text{Sb},\text{Bi})_2$ samples at 500-800 K, its carrier mobility and thermal conductivity are less different from that of unadded samples. As the temperature rises, the influence of energy filter effect will gradually weaken, so is the slight decrease in Seebeck coefficient simply due to the increase in carrier concentration? Otherwise, what are the main factors responsible for the improvement of zT at high temperatures?

Reviewer #2 (Remarks to the Author):

This manuscript reported a recorded high thermoelectric performance of Mg_3Sb_2 with Nb nano-additions. The material transport properties as well as the single-leg device efficiency are reproducible. Based on these, I can recommend a publication in this journal. However, I would suggest the authors to include more analyses and comparisons for supporting their claims about energy filtering.

1. I would suggest a Pisarenko plot, particularly at low temperatures, to demonstrate the existence of energy filtering.

2. Since the authors reported a recorded high PF, it is better to compare the Hall mobility, resistivity and Seebeck coefficient with the state-of-the-art Mg_3Sb_2 in Figure 1. In this way, the readers can clearly find what kind of transport properties are mainly improved due to energy filtering.

3. Why Nb or Ta is so special and effective for inducing energy filtering in Mg_3Sb_2 . The authors are suggested to provide more discussion or calculation on this issue.

4. Why Nb is superior to Ta for reducing lattice thermal conductivity?
5. I would suggest a predicted efficiency plot in Figure 6d.
6. The authors used the phrases of “positive or negative effects” many times, these effects should be clearly defined.

Reviewer #3 (Remarks to the Author):

In this work, the authors reported the effects of additional metallic elements on thermoelectric properties of $\text{Mg}_3(\text{SbBi})_2$ -based compounds. In previous studies, the higher electronic properties with Nb addition, larger grains were both already reported. However, the mobility value around room temperature is even higher than these reports and authors added some new insight into this improvement. Since Mg_3Sb_2 is a hot topic in the field of thermoelectrics and the results of this work are important for the development of thermoelectric devices, I can recommend publishing this work in nature communications. However, some additional explanations would be required before acceptance.

1. Why Nb is chosen as the first example among all the possible elements? If the enhancement of the performance originated from lower grain boundary resistance and metallic inclusions, can we expect similar effects from most of the metals? Why Ta have less effective compared to Nb? Are there any indicators to identify the effective (beneficial) additives?
2. Authors used Nb powder and Ta powder as additives. Is there any difference if the starting elements are different form such as smaller mesh or grains.
3. Page 3, line 93, “However, a slight decrease in S was observed at low temperatures in the $x = 0.2$ sample”. Do authors have any speculation of this change or is it caused by the measurement set up?
4. Page 3 line 111, could you please cite the previous report that authors are referring to?
5. Based on their statement, I assume Nb is not substituting Mg site, and exists as the secondary phase at the grain boundary. Does that statement contradict the nH changes depending on Nb content x? (Line 118, Line 158) Considering the amount of Nb added ($x=0.1, 0.2$), isn't the carrier concentration change too small if it is depending on the Nb content x?
6. Regarding the statement on Line 161, grain boundary growth is hindered by Nb inclusions, is seems different from the previous Nb added study on microstructure. (ref 22) Are there any differences in the condition or mechanisms?
7. What do author think about the mean free path of phonon and electron? Is there large enough difference between these two so that Nb nano inclusions (wide range of size of 10 – 1000nm) can selectively scatter only phonons without decreasing carrier mobility?
8. The energy filtering picture is not convincing. Is the difference in Seebeck coefficient large enough considering uncertainty and sample to sample difference etc.? How much improvement is coming from reduced grain boundary resistance and could it be analyzed separately from energy filtering effect? Further analysis would be required to use energy filtering story to explain the change in electrical properties.

Sep. 1st, 2023

Dear Respected Editor and reviewers,

Thank you very much for your time and efforts in handling our manuscript (Research Article, No. NCOMMS-23-30931). We greatly appreciate the reviewers' valuable comments and suggestions. Following are our responses to the reviewers' comments, based on which we have thoroughly revised our manuscript. We also prepared the samples with more compositions and did the related measurement as suggested. We hope the revised manuscript will address the concerns of the reviewers and meet the requirements of your esteemed journal *Nature Communications*.

Answers to reviewers:

Reviewer #1:

General Comment:

The paper presents a comprehensive investigation into the impact of metallic inclusions (specifically, Nb and Ta) on the thermoelectric properties of n-type $\text{Mg}_3(\text{Sb,Bi})_2$ -based material. The authors identified the presence of a Nb-rich second phase at grain boundaries, which leads to grain growth inhibition and a substantial enhancement of electric transport properties, attributed to reduced interface potential barriers. Impressively, this study achieves a zT of 0.8 at room temperature and a notable peak zT value of 2.0 at 798 K. Moreover, a single leg device demonstrates a remarkable energy conversion efficiency of 14% across a temperature differential of 470 K. The research results are robust, and the manuscript is well-organized. However, certain aspects and major concerns have to be addressed prior to consideration for publication in Nature communications.

Response: Thanks for your valuable comments.

Major comments:

1. Wang et al. published a similar study on the performance optimization of $\text{Mg}_3(\text{Sb,Bi})_2$ thermoelectrics through Mo addition using the same ball-milling and with SPS method (Adv. Energy Mater. 2023, 2301667). Their results also demonstrated the increased electron transport properties and decreased lattice thermal conductivities in Mo-added samples. Meantime, many studies have focused on the doping of transition metal(alloys) in the $\text{Mg}_3(\text{Sb,Bi})_2$ system. The authors need to conduct further literature review and clarify the novelty of this work.

Response:

Thank you for your suggestion. Actually, we have noticed this article you mentioned above in recent days. Since this article was first published on July 27, and our article was submitted on July 24, we had no chance to study this article before we submitted

our work. The article you mentioned is a very valuable work, also showing that the $\text{Mg}_3(\text{Sb,Bi})_2$ system is competitive in thermoelectric materials and has huge application potential.

Compared with the work for Mo addition, a higher zT of 2.04 at 798 K and 0.8 at 300 K is obtained in our work than 1.84 at 723 K and 0.4 at 300 K in the Mo-added work. Moreover, the two works show different mechanisms for improvement. The σ increases with increasing the amount of Nb in our work. However, for a series of Mo-added samples, the σ will first increase and then decrease with the added amount (Figure R1a), and the highest σ is obtained when $x=0.04$.

The above article proved that Mo addition in the samples contributed to increasing average grain size via GB segregation, thereby obtaining the improved μ and σ . However, our work is different in several aspects. In our work, the use of higher sintering temperature would theoretically result in larger grains, but the reduced grains were observed, which is contrary to the results of Mo addition as shown in Figure R1b. Therefore, we propose the fast transport channels for carriers via Nb addition and discuss the reduction of the interfacial barriers. It can be inferred that the ability to reduce the interfacial barrier through Nb should be superior to Mo. We have also cited the above article in the manuscript and given the relevant discussion.

[Redacted]

Figure R1. a) Temperature-dependent electrical conductivity of the $\text{Mg}_{3.2-x}\text{Mo}_x\text{Sb}_y\text{Bi}_{1.99-y}\text{Te}_{0.01}$ samples. b) Scanning electron microscope (SEM) images from polished and HNO_3 etched surface of $\text{Mg}_{3.2-x}\text{Mo}_x\text{Sb}_y\text{Bi}_{1.99-y}\text{Te}_{0.01}$ samples. (*Adv. Energy Mater.* **2023**, 2301667)

Revision:

Recently, it has been reported that the Mo addition also increases the σ of $\text{Mg}_3(\text{Sb,Bi})_2$ -based materials³⁸, indicating the great potential of other transition metals. However, the enhanced σ by Mo addition is mainly due to the grain boundary effect, different from those analyzed above for the Nb or Ta inclusions.

2. This work mainly studied $\text{Mg}_3\text{Sb}_{1.5}\text{Bi}_{0.49}\text{Te}_{0.01}$ based samples. However, $x\text{Nb}/\text{Mg}_3(\text{Sb},\text{Bi})_2$ and similar expressions were used throughout the manuscript, which may mislead readers and to some extent was over claiming.

Response:

Thank you for your comments. To avoid misunderstanding, we have revised the expression to $x\text{Nb}/\text{Mg}_3\text{Sb}_{1.5}\text{Bi}_{0.49}\text{Te}_{0.01}$.

Revision: $x\text{Nb}/\text{Mg}_3\text{Sb}_{1.5}\text{Bi}_{0.49}\text{Te}_{0.01}$

3. Figure 1b and the inset of Figure 1d present an inconsistency between the trend of the Seebeck coefficient and carrier concentration. Given their leading roles in thermoelectric behavior, it is essential to address this inconsistency comprehensively. The introduction of a Pisarenko line encompassing all compositions, along with an explanatory analysis is required.

Response:

Thank you for your comments. We have calculated the Pisarenko plot, showing the S slightly deviates from the Pisarenko plot and is consistent with the case of energy filtering (*Adv. Funct. Mater.* **2020**, *30*, 1901789). As shown in **Figure R2**, for a fixed density-of-state effective mass (m^*) of $1.35 m_e$, the data points of Nb-added samples sintered at 1073 K are very close to the curve, but deviate upward slightly with increasing the amount of Nb. The point corresponding to the Ta-added sample also shows a similar deviation. Assuming that the Nb or Ta inclusions do not enter the crystal lattice as dopants, the deviation should be mainly attributed to the introduced metallic inclusions. The metallic inclusion helps to reduce the interfacial barriers induced by high-resistivity grain boundary, increasing the contribution of carriers to the S by the fast transport channel effect of carriers.

However, both the samples sintered at 923 K are fitted well with a m^* of $\sim 1.85m_e$ almost insensitive to the Nb addition. The m^* reduces from $\sim 1.85 m_e$ to $1.35 m_e$ with increasing sintering temperature, which may be attributed to the massive defects generated in $\text{Mg}_3\text{Sb}_{1.5}\text{Bi}_{0.5}$ -based materials at higher sintering temperatures (*Adv. Mater.* **2023**, *35*, 2209119). Since the diffusion at the interface sintered at 923 K is not as sufficient as that sintered at 1073 K, the interface may have a weaker contact, leading to less reduction of the interfacial barriers. As a result, there is no significant deviation shown in the curve.

Figure R2. Pisarenko plots show the Seebeck coefficients as a function of carrier concentration at 300 K.

Revision:

Mostly, the increase of n_H leads to a decreased S . But there is an inconsistency between the trend of the S and n_H for the 1073 K-sintered samples. This may be attributed to the reduced interfacial barriers allowing more high-energy carriers to contribute to the S (this will be anatomized later). Therefore, a higher S is still observed in $x = 0.1$ sample with an increased n_H .

The contribution of high-energy carriers to transport properties is enhanced, accounting for a higher S . In the case for Nb-added samples, this modified interfacial barrier turns out to optimize σ without sacrificing S , thereby leading to enhanced PF. The S may be a synergistic result of the above effect and increased n_H . The Pisarenko plots shown in Supplementary Fig. 17 exhibit the contribution from the modified interfacial barrier to S , where the data points slightly deviate upward.

(Supporting Information) For a fixed density-of-state effective mass (m^*) of $1.35 m_e$, the data points of Nb-added samples sintered at 1073 K are very close to the curve, but deviate upward slightly with increasing the amount of Nb. The point corresponding to the Ta-added sample also shows a similar deviation. However, both the samples sintered at 923 K are fitted well with a m^* of $\sim 1.85 m_e$ regardless of the addition of Nb. The m^* reduces from $\sim 1.85 m_e$ to $1.35 m_e$ with increasing the sintering temperature, which may be attributed to the massive defects generated in $Mg_3Sb_{1.5}Bi_{0.5}$ -based materials at higher sintering temperatures. Since the diffusion at the interface sintered at 923 K is not as sufficient as that sintered at 1073 K, the interface may have a weaker contact, leading to less reduction of the interfacial barriers. As a result, there is no significant deviation shown in the curve.

4. Lines 97-99: "Clearly,...optimization of electrical transport performance at low temperatures', Only two compositions of Nb-added samples, i.e.0.1 Nb and 0.2 Nb, were prepared in this study, thus, the statement here was not solidly supported.

Generally, this study delivered only three compositions of Nb/Ta-added samples, which was not as convincing as one may expect from a comprehensive study published on Journals such as Nature Communications.

Response:

Thank you for your comments. To further support the statement in the manuscript, we prepared the samples with $x=0.05$ and $x=0.15$ and then measured the TE properties, as shown in **Figure R3**. The σ monotonically increased upon increasing the amount of Nb. Owing to no significant deterioration of S , all the Nb-added samples showed optimization of PF , especially at low temperatures.

Since we would like to emphasize here that Nb affects optimizing electrical properties rather than the specific effect of the amount of Nb, the above supplementary compositions have been added to the Supporting Information.

Figure R3. Temperature dependence of the (a) electrical conductivity, (b) Seebeck coefficient, (c) power factor, (d) total thermal conductivity, (e) lattice thermal conductivity and (f) zT of $x\text{Nb}/\text{Mg}_3\text{Sb}_{1.5}\text{Bi}_{0.49}\text{Te}_{0.01}$ samples. The insets in (a) and (e) show the electrical conductivity and the lattice thermal conductivity depending on x at 300 K, respectively.

Revision:

The electrical properties of the sample with $x = 0.05$ and 0.15 are also shown in **Supplementary Fig. 2**.

5. Discussion in the paragraph from line 100 was not very rigorous and lacks supports from literature studies. E.g. "In fact, the changes in n_H and μ_H cannot be ascribed to the variable point defects (e.g. Mg vacancies) as thought conventionally." Refs should be

cited and explanations should be included." This might be ascribed to the Sb deficiency in the matrix caused by the formation of the Nb₃Sb phase." Links to related results should be made and no clear evidence on Sb deficiency in the matrix was available in the current manuscript and SI.

Response:

#Regarding the discussion in the paragraph from line 100.

Thank you for your comments. Considering that the carrier scattering induced by point defects is usually more intensive at high temperatures, the high-temperature electrical properties would demonstrate a larger difference. However, in our work, the mobility is almost constant at high temperatures, but shows significant differences near room temperature. Therefore, the grain boundary effect is regarded as the dominant factor. To make the expression more clear, we have cited the references and revised the two sentences.

Revision:

In fact, the changes in n_H and μ_H cannot be solely ascribed to the variable point defects (e.g. Mg vacancies²⁹ or doping atoms²⁷) as thought conventionally, because the carrier scattering induced by point defects is usually more intense in the high-temperature range, which significantly affects the electrical properties at high temperatures.

#Regarding the Sb deficiency.

Thank you for your valuable comments. We apologize for the inappropriate use of the term "Sb deficiency", which may cause misunderstanding.

In the initial version of our manuscript, based on the results of the secondary phase of Nb₃Sb observed by HRTEM, we proposed the presumption to explain the enhancement of n_H . Since it is confirmed that Nb atoms do not enter the lattice for substitution, we presume that the Sb atoms in Nb₃Sb can only come from the matrix. To verify this presumption, we have supplemented the EPMA analysis (Figure R4) and added it to the SI. Fortunately, the quantitative analysis of elements by EPMA can support the above conclusion. Multiple grains were randomly selected in each sample to detect the atomic ratio of each element. For the samples sintered at 1073 K, the atomic ratio of Sb/Mg in the matrix of $x = 0.1$ sample is 0.595, slightly lower than 0.650 of $x = 0$ sample. It indicates that a small amount of Sb in the matrix is consumed to form Nb₃Sb. However, due to this amount being small, the expression of "a small amount of Sb atoms being consumed" or "the proportion of Sb decreases" may be more appropriate. In addition, the Bi-rich grain boundary phases with similar compositions were observed in both $x = 0$ and 0.1 samples, which was consistent with the previous work (*Adv. Energy Mater.* **2023**, 2301667).

Figure R4. Back-scattering images from the polished surface of $x\text{Nb}/\text{Mg}_3\text{Sb}_{1.5}\text{Bi}_{0.49}\text{Te}_{0.01}$ ($x = 0$ and 0.01) sintered at 1073 K and corresponding point composition estimated by EPMA. The presence of Nb signals may be due to the residue of Nb inclusions during polishing or Nb inclusions whose size is less than the probe limit of $1\mu\text{m}$.

Revision:

Besides, due to the formation of the Nb_3Sb phase near the interface, a small amount of Sb atoms in the matrix would be inevitably consumed by Nb. As shown in Supplementary Fig. 10, the results of electron probe microanalysis (EPMA) confirmed that the atomic ratio of Sb/Mg in the matrix of $x = 0.1$ sample is lower than that of the unadded sample. The decreased Sb/Mg ratio may suppress the formation of Mg vacancies, which is beneficial to increasing the n_{H} and hence the σ of $\text{Mg}_3(\text{Sb},\text{Bi})_2$.

6. Lines 208-210: similar to above, how does the Sb deficiency affect n_{H} and σ ? Are there any reported studies? Explanations should be made in this part.

Response:

We apologize for the inappropriate use of the term "Sb deficiency", which is more appropriate to express as "a small amount of Sb atoms being consumed" or "the proportion of Sb decreases". The above EPMA results can well support our statement that the Sb atoms from the matrix were consumed. As shown in Figure R4, the ratio of Sb/Mg in the matrix is decreased for the $x = 0.1$ sample. Therefore, it is very possible to suppress the formation of Mg vacancies due to the decreased proportion of Sb. Therefore, when Mg vacancies as electron killers are reduced, the electron concentration, that is, the carrier concentration, would be increased, resulting in an enhancement in σ .

Revision:

Besides, due to the formation of the Nb₃Sb phase near the interface, a small amount of Sb atoms in the matrix would be inevitably consumed by Nb. As shown in **Supplementary Fig. 10**, the results of electron probe microanalysis (EPMA) confirmed that the atomic ratio of Sb/Mg in the matrix of $x = 0.1$ sample is lower than that of the unadded sample. The decreased Sb/Mg ratio may suppress the formation of Mg vacancies, which is beneficial to increasing the n_H and hence the σ of Mg₃(Sb,Bi)₂.

7. Since a substantial portion of a Nb-rich second phase is found at grain boundaries, as confirmed by TEM, SEM, and XRD analyses, it is imperative to provide the thermal transport properties of this Nb-rich compound.

Response:

Thank you for your comments. The Nb-rich phase can be approximated as pure Nb metal particles, of which the thermal conductivity is $53.7 \text{ W m}^{-1} \text{ K}^{-1}$. The presence of the Nb₃Sb phase at the interface was observed by HRTEM. The secondary phase can be proved to exist stably by the Nb-Sb binary phase diagram (**Figure R5a**). However, it must be emphasized that the amount of the Nb₃Sb phase is very small. From the XRD pattern in **Figure S4**, only the peaks of Nb and no peaks of the Nb₃Sb phase can be observed. For the powder sample ground after sintering, the Nb phase can be obviously detected by HAADF-STEM instead of the Nb₃Sb phase, as shown in **Figure S6**. To characterize the morphology of the interface, the HAADF experiments were carried out on the bulk samples via ion thinning. In **Figure 3a**, the Nb₃Sb phase was found in region H at the interface. However, the EDS analysis of the central region (F) of Nb inclusions showed that the proportion of Nb atoms was as high as 86%, much higher than that in the Nb₃Sb phase. The weak signal of Mg and Sb may come from the influence of background. Therefore, it can be considered that the inclusions have a core-shell-like structure with Nb as the main body and a small amount of Nb₃Sb as the coating layer, as demonstrated in **Figure R5b**. For the discussion of performance, the thermal conductivity of Nb metal should be considered.

Figure R5. a) The Nb-Sb binary phase diagram³⁰. b) Schematic illustration of the structure of Nb inclusion.

Revision:

The Nb-Sb binary phase diagram can serve as evidence for the presence of stable Nb₃Sb phases (Supplementary Fig. 8a)³⁰.

The inclusions perhaps have a core-shell-like structure with Nb as the main body and a small amount of Nb₃Sb as the coating layer, as demonstrated in Supplementary Fig. 8b.

8. More experiments and more compositions are suggested to evaluate κ and κ_L . Why the κ and κ_L values contrasted remarkably between 0.1Nb/Mg₃Sb_{1.5}Bi_{0.49}Te_{0.01} and 0.2Nb/Mg₃Sb_{1.5}Bi_{0.49}Te_{0.01} but negligibly between 0.1Nb/Mg₃Sb_{1.5}Bi_{0.49}Te_{0.01} and Mg₃Sb_{1.5}Bi_{0.49}Te_{0.01}? What are the optimized compositions?

Response:

Thank you for your comments. We have prepared the samples with $x=0.05$ and $x=0.15$ and then measured the thermal transport properties, as shown in Figure R6. The κ and κ_L decrease first and then increase upon the addition of Nb. The κ reaches the lowest when $x = 0.05$ and the κ_L reaches the lowest when $x = 0.1$, benefiting from the strong scattering of phonons by the nano-inclusions. However, with the addition increasing, the contribution of the high thermal conductivity of the Nb phase will exceed the reduction of the thermal conductivity by phonon scattering, resulting in increased thermal conductivity.

Figure R6. Temperature dependence of the (a) total thermal conductivity and (b) lattice thermal conductivity of $x\text{Nb}/\text{Mg}_3\text{Sb}_{1.5}\text{Bi}_{0.49}\text{Te}_{0.01}$ samples. The inset in (b) shows the lattice thermal conductivity depending on x at 300 K.

Revision:

The κ and κ_L decrease first and then increase upon the addition of Nb. When the contribution of the high κ of Nb phase ($53.7 \text{ W m}^{-1} \text{ K}^{-1}$) exceeds the reduction on κ by phonon scattering, it results in increased κ and κ_L .

9. Are there any links between the energy-filtering effect in paragraph (lines 278-290) and previous discussion & microstructures?

Response:

Thank you for your comments. Different from the generally mentioned energy-filtering effect via introducing barriers, the tailored energy-filtering effect we discussed in this work is caused by the reduced interfacial barrier when considering the introduced barriers by grain boundaries. The previous work (*Energy Environ. Sci.* **2009**, 2, 466) has shown that grain boundaries can negatively affect mobility, but also possibly play a positive role through energy filtering. As shown in **Figure R7**, considering that the barrier introduced by grain boundaries is located at E_{b1} (green box), the electrons with negative Seebeck distribution and a part of electrons with positive Seebeck distribution are scattered by the interfacial barriers. After adding metallic inclusions, the interfacial barrier would be reduced from E_{b1} to E_{b2} (red box). In this case, more electrons with positive Seebeck distribution are able to pass through the barrier, thus increasing the Seebeck coefficient.

Consequently, it is believed that the addition of metallic inclusions and the reduced interfacial barriers have a favorable regulation of the existing energy-filtering effect. To avoid misunderstanding, we have revised the manuscript and replaced the expression of “tailored energy filtering effect” with “modified interfacial barrier”.

[Redacted]

Figure R7. Calculated normalized Seebeck distribution versus energy. Low energy electrons reduce the total Seebeck coefficient. (*Energy Environ. Sci.* **2009**, 2, 466).

Revision:

Meanwhile, the S could be obviously enhanced in the $\text{Mg}_3(\text{Sb,Bi})_2$ with Ta addition, indicating that these built-in metallic nano-inclusions effectively modified the interfacial barriers and enhanced the contribution from high-energy electrons to transport properties.

By incorporating Ta inclusions, the interfacial barrier is reduced from E_{b1} to E_{b2} , allowing more high-energy carriers to pass through. The contribution of high-energy carriers to transport properties is enhanced, exhibiting a higher S . In the case of Nb-added samples, this modified interfacial barrier turns out to optimize σ without sacrificing S , thereby leading to enhanced PF.

10. In conclusions: "Combined with the effective phonon scattering effect caused by the metal inclusions, the lattice thermal conductivity was also obviously diminished." There seems no rigorous evidence on the effective phonon scattering effect caused by the metal inclusions (e.g. Figure 4a). The 0.1Nb and 0.2Nb samples depicted different trends on scattering phonons.

Response:

We apologize for the lack of rigor in the original statement. As mentioned in the reply to comment 8, the κ and κ_L decrease first and then increase upon the addition of Nb (Figure R6). When the addition amount is small, the metallic inclusions can play the role of scattering phonons. However, with the addition increasing, the contribution of the high thermal conductivity of the metal phase will exceed the reduction of the thermal conductivity by phonon scattering, resulting in increased thermal conductivity. And we have revised this sentence in the manuscript.

Revision:

However, the κ significantly increases when the amount of Nb increases to more than 0.1; higher fractions of thermally conducting Nb phases are more likely to affect the κ through its intrinsic nature. When the contribution of the high κ of Nb phase ($53.7 \text{ W m}^{-1} \text{ K}^{-1}$) exceeds the reduction on κ by phonon scattering, it results in increased κ and κL .

Combined with the effective phonon scattering effect caused by a small amount of metallic inclusions, the lattice thermal conductivity was also obviously diminished.

11. A central innovation of this work revolves around the reduction of potential barriers between the grains due to the metallic inclusions. To bolster the clarity and significance of the findings, a fabrication of Nb contact layer on the optimized $\text{Mg}_3(\text{Sb,Bi})_2$ materials with an in-depth investigation of the transport properties of the Nb/ $\text{Mg}_3(\text{Sb,Bi})_2$ interface is recommended.

Response:

We appreciate your professional suggestion. It is a valuable idea for $\text{Mg}_3(\text{Sb,Bi})_2$ materials. Before we were ready to experiment, we found the two articles investigating the Nb/ Mg_3SbBi interface layer (*J. Inorg. Mater.* **2023**, *38*, 931 and *Energy Environ. Sci.* **2022**, *15*, 3265). Fu and Hu et al. have proved that the Nb/ Mg_3SbBi interface has good contact and there is a certain diffusion bonding, as shown in Figure R8a and R9a. The contact resistivity of Nb/ Mg_3SbBi is 9.7 or $12.9 \mu\text{W}\cdot\text{cm}^2$, which is a competitive value in the total Mg_3Sb_2 -based TE devices (Figure R8b and R9b). In addition, there is no obvious elemental diffusion at the interface after aging (Figure R9c and R9d), indicating that the Nb/ Mg_3SbBi interface has excellent thermal stability and low contact resistivity (Figure R8c).

We have cited these two articles in the manuscript. And these articles also proved the value and importance of your suggestion. Their work showed that Nb is predicted as a contact layer, which is expected to be used in the TE device preparation in the future.

[Redacted]

Figure R8. a) EDS line scanning results of the $\text{Mg}_{3.2}\text{Bi}_{0.996}\text{SbSe}_{0.004}/\text{Nb}$ junction before and after aging. The inset is a photo of the tested sample. (b) Measured contact resistivity (ρ_c) of the as-sintered $\text{Mg}_{3.2}\text{Bi}_{0.996}\text{SbSe}_{0.004}$ junction in comparison with

literature results. (c) ρ_c as a function of thermal aging temperature and time. (*Energy Environ. Sci.* **2022**, *15*, 3265)

Figure R9. a) BSE image of Nb/Mg₃SbBi sample before aging. b) Interface resistivity of Nb/Mg₃SbBi and comparison with literature. Backscatter (left) and line sweep (right) results of Nb/Mg₃SbBi interface aged at 525°C (c) 70 h and (d) 170 h.

Revision:

Moreover, the investigation of the Nb/Mg₃SbBi interface also confirms that Nb is in good contact with the matrix without macroscopic secondary phase.^{32,33}

12. As the author state that Ta was expected to exhibit similar behaviors. In the inset of Figure 5a, the high temperature Seebeck coefficient of Ta0.1-added sample was improved significantly while the addition of Nb didn't show a similar effect. And also, in Figure S13, the addition of Nb and Ta have opposite effects on the lattice thermal conductivity. The author should clarify and have in depth discussion on it.

Response:

Thank you for your comments. As we mentioned in comment 9, reducing the interfacial barrier is beneficial for increasing *S*. Meanwhile, the *S* would exhibit a decrease when *n_H* increases. Therefore, the *S* may be a synergistic result of the above effect and

increased n_H . Since the n_H of Ta-added samples only increased slightly, the increase in S due to the reduction of interfacial barrier is more obvious. Different from the case of Nb addition, no Ta-Sb binary phase diagram is found. It is impossible to form a Ta-rich secondary phase similar to Nb_3Sb . This may be the reason why the n_H of the Ta-added sample has a less increase than that of the Nb-added sample.

The higher thermal conductivity of the Ta-added sample may be due to the following two reasons. On the one hand, due to the thermal conductivity of Ta ($57.5 \text{ W m}^{-1} \text{ K}^{-1}$) being slightly higher than that of Nb, the contribution of Ta inclusions to the total thermal conductivity is higher. On the other hand, by comparing Supplementary Fig. 5 and 15, it can be seen that the size of Ta inclusions in the sintered bulk is larger than that of Nb, which results in a weaker scattering on phonons than that of the small-size inclusions. To avoid confusion, we have added some in-depth discussion to the manuscript.

Supplementary Figure 5. EDS mapping of $0.1\text{Nb}/\text{Mg}_3\text{Sb}_{1.5}\text{Bi}_{0.49}\text{Te}_{0.01}$ sintered at 1073 K on the fractured surface.

Supplementary Figure 15. EDS mapping of 0.1Ta/Mg₃Sb_{1.5}Bi_{0.49}Te_{0.01} sintered at 1073 K on the fractured surface.

Revision:

This may be because no Ta-rich secondary phases similar to Nb₃Sb were formed under the present experimental conditions. In fact, no Ta-Sb binary phase diagram was reported. That is why the Ta addition did not affect the carrier concentration as in the case of the Nb-added samples.

However, due to the higher κ ($57.5 \text{ W m}^{-1} \text{ K}^{-1}$) as well as weaker scattering originating from the larger size of Ta inclusions, the effect of Ta inclusions on reducing κ is limited (Supplementary Fig. 18).

13. It is suggested to predict the performance of the single leg device based on the thermoelectric properties of the optimized materials. A comparison between predicted and measured results would not only elucidate the quality of device fabrication but also provide insights into potential measurement errors.

Response:

Thank you for your suggestion. We have simulated the theoretical conversion efficiency by the COMSOL Multiphysics software and added the results to the SI.

By comparing the predicted (Figure R9) and measured parameters (Supplementary Fig. 19), it can be identified that the measured output power is slightly lower than the predicted value and the heat flow is higher. The difference in output power mainly stems from the contact resistance. The excessive heat flow results from heat leakage due to technical limitations. The heating table area (10*10 mm) of the Mini-PEM device is larger than the sample size (4.5*5 mm), leading to heat being more easily radiated from the hot side to the cold side. As a result, the Q_c measured by the heat flow meter on the cold side is larger than the Q_c actually flowing through the sample. The predicted efficiency of single-leg device is 18%. It is believed that the measured efficiency is

expected to be improved by further optimizing the contact layer and controlling the influence of thermal radiation in the measurement.

Figure R9. The predicted (a) voltage, (b) output power (c) output heat and (d) efficiency as a function of the current under different hot-side temperatures for the single leg based on the zT value of 0.1Nb/Mg₃Sb_{1.5}Bi_{0.49}Te_{0.01}. The cold-side temperature is 300 K.

Figure S18. The measured (a) voltage, (b) output power (c) output heat and (d) efficiency as a function of the current under different hot-side temperatures for the single leg 1 of 0.1Nb/Mg₃Sb_{1.5}Bi_{0.49}Te_{0.01}. The cold-side temperature is 295 K.

Revision:

The theoretical efficiency of the single-leg device reaches 18% as shown in **Figure S22**,

but both the measured output power and heat flow deviate from the predicted value, indicating more efforts are needed to optimize the contact layer and control the thermal radiation.

14. As for the thermoelectric performance of 0.1Nb/Mg₃(Sb,Bi)₂ samples at 500-800 K, its carrier mobility and thermal conductivity are less different from that of unadded samples. As the temperature rises, the influence of energy filter effect will gradually weaken, so is the slight decrease in Seebeck coefficient simply due to the increase in carrier concentration? Otherwise, what are the main factors responsible for the improvement of zT at high temperatures?

Response:

Thank you for your comments. In our opinion, the decrease in S is due to the increased n_H . Moreover, the effect of n_H on S becomes clearer with increasing temperature. We believe that the improvement of zT value at high temperatures mainly stems from the slight decrease in κ (from 0.745 to 0.715 W m⁻¹ K⁻¹, a 4% decrease) and the increase in σ (from 180 to 211 S cm⁻¹, a 17% increase). The σ is contributed by the increased n_H . Since the zT value of the unadded sample has been optimized to a high level through high-temperature sintering, there is an impressive improvement of zT when a small optimization of TE parameters can be obtained.

We are sorry that there is little discussion of high temperature properties in the manuscript. To make the results more clearly, we have added some discussion on the improvement of zT at high temperatures to the manuscript.

Revision:

Meanwhile, the slight decrease in S mainly stems from the increased n_H . Similarly, a more pronounced decrease in S is observed due to higher n_H in the $x = 0.2$ sample. Nevertheless, the augmented electron concentration donated by Nb still guarantees higher σ (a 17% increase at 798 K), leading to enhanced PF at high temperatures.

Reviewer #2 (Remarks to the Author):

This manuscript reported a recorded high thermoelectric performance of Mg₃Sb₂ with Nb nano-additions. The material transport properties as well as the single-leg device efficiency are reproducible. Based on these, I can recommend a publication in this journal, However, I would suggest the authors to include more analyses and comparisons for supporting their claims about energy filtering.

Response: Thanks for your positive and valuable comments.

1. I would suggest a Pisarenko plot, particularly at low temperatures, to demonstrate the existence of energy filtering.

Response:

Thank you for your comments. We have calculated the Pisarenko plot, showing the S slightly deviates from the Pisarenko plot and is consistent with the case of energy filtering (*Adv. Funct. Mater.* **2020**, *30*, 1901789). As shown in **Figure R2**, for a fixed density-of-state effective mass (m^*) of $1.35 m_e$, the data points of Nb-added samples sintered at 1073 K are very close to the curve, but deviate upward slightly with increasing the amount of Nb. The point corresponding to the Ta-added sample also shows a similar deviation. Assuming that the Nb or Ta inclusions do not enter the crystal lattice as dopants, the deviation should be mainly attributed to the introduced metallic inclusions. The metallic inclusion helps to reduce the interfacial barriers induced by high-resistivity grain boundary, increasing the contribution of carriers to the S by the fast transport channel effect of carriers.

However, both the samples sintered at 923 K are fitted well with a m^* of $\sim 1.85 m_e$ regardless of the addition of Nb. The m^* reduces from $\sim 1.85 m_e$ to $1.35 m_e$ with increasing the sintering temperature, which may be attributed to the massive defects generated in $\text{Mg}_3\text{Sb}_{1.5}\text{Bi}_{0.5}$ -based materials at higher sintering temperatures (*Adv. Mater.* **2023**, *35*, 2209119). Since the diffusion at the interface sintered at 923 K is not as sufficient as that sintered at 1073 K, the interface may have a weaker contact, leading to less reduction of the interfacial barriers. As a result, there is no significant deviation shown in the curve.

Figure R2. Pisarenko plots show the Seebeck coefficients as a function of carrier concentration at 300 K.

Revision:

Mostly, the increase of n_H leads to a decreased S . But there is an inconsistency between the trend of the S and n_H for the 1073 K-sintered samples. This may be attributed to the reduced interfacial barriers allowing more high-energy carriers to contribute to the S (this will be anatomized later). Therefore, a higher S is still observed in $x = 0.1$ sample with an increased n_H .

The contribution of high-energy carriers to transport properties is enhanced, accounting for a higher S . In the case for Nb-added samples, this modified interfacial barrier turns out to optimize σ without sacrificing S , thereby leading to enhanced PF. The S may be a synergistic result of the above effect and increased n_H . The Pisarenko plots shown in Supplementary Fig. 17 exhibit the contribution from the modified interfacial barrier to S , where the data points slightly deviate upward.

(Supporting Information) For a fixed density-of-state effective mass (m^*) of $1.35 m_e$, the data points of Nb-added samples sintered at 1073 K are very close to the curve, but deviate upward slightly with increasing the amount of Nb. The point corresponding to the Ta-added sample also shows a similar deviation. However, both the samples sintered at 923 K are fitted well with a m^* of $\sim 1.85 m_e$ regardless of the addition of Nb. The m^* reduces from $\sim 1.85 m_e$ to $1.35 m_e$ with increasing the sintering temperature, which may be attributed to the massive defects generated in $Mg_3Sb_{1.5}Bi_{0.5}$ -based materials at higher sintering temperatures. Since the diffusion at the interface sintered at 923 K is not as sufficient as that sintered at 1073 K, the interface may have a weaker contact, leading to less reduction of the interfacial barriers. As a result, there is no significant deviation shown in the curve.

2. Since the authors reported a recorded high PF, it is better to compare the Hall mobility, resistivity and Seebeck coefficient with the state-of-the-art Mg_3Sb_2 in Figure 1. In this way, the readers can clearly find what kind of transport properties are mainly improved due to energy filtering.

Response:

Thank you for your comments. We compared the Hall mobility, conductivity and Seebeck coefficient with the state-of-the-art Mg_3Sb_2 and added the results to the SI, as shown in Figure R10. It can be seen that the Seebeck coefficient in this work is higher than that in other works. Meanwhile, the conductivity and Hall mobility still remain at a high level, which leads to a record-high PF.

Figure R10. The comparison of (a) electrical conductivity, (b) Seebeck coefficient, and (c) Hall mobility in this work with previously reported results^{15,21,27,28}.

Revision:

Essentially, the S contributes more to the high PF while the σ is comparable to the values reported in other literatures, as demonstrated in Supplementary Fig. 1.

3. Why Nb or Ta is so special and effective for inducing energy filtering in Mg₃Sb₂. The authors are suggested to provide more discussion or calculation on this issue.

Response:

Thank you for your comments. First, before we explain the particularity of Nb and Ta, we would like to clarify the so-called tailored energy-filtering effect in this work.

Different from the generally mentioned energy-filtering effect via introducing barriers, the tailored energy-filtering effect we discussed in this work is caused by the reduced interfacial barrier when considering the introduced barriers by grain boundaries. The previous work (*Energy Environ. Sci.* **2009**, 2, 466) has shown that grain boundaries can provide an interfacial barrier and negatively affect mobility, but also possibly play a positive role through energy filtering. After adding metallic inclusions, the interfacial barrier would be reduced. In this case, more electrons with positive Seebeck distribution are able to pass through the barrier, thus increasing the Seebeck coefficient. To clarify the discussion, we have revised the manuscript and replaced the expression of “tailored energy filtering effect” with “modified interfacial barrier”.

Regarding the particularity of Nb and Ta, it may be due to the following three aspects. First, the large atomic size mismatch and the low solid solubility of Nb and Ta make them not substituted at the lattice site. The second is that Nb and Ta form a secondary phase similar to metal on the grain boundary. The third is the formation of an intimate contact between the metallic inclusions and the matrix. Owing to the high electrical conductivity of the metal phase and the great bonding at the interfaces, the interfacial barrier is effectively reduced, resulting in the improvement of TE performance.

Based on the above discussion, it is believed that a similar effect is possible by adding other transition metals, which is worth further investigating the effectiveness of more elements in the future.

4. Why Nb is superior to Ta for reducing lattice thermal conductivity?

Response:

Thank you for your comments. The higher thermal conductivity of the Ta-added sample may be due to the following two reasons. On the one hand, due to the thermal conductivity of Ta ($57.5 \text{ W m}^{-1} \text{ K}^{-1}$) being slightly higher than that of Nb, the contribution of Ta inclusions to the total thermal conductivity is higher. On the other hand, by comparing **Supplementary Fig. 5** and **15**, it can be seen that the size of Ta inclusions in the sintered bulk is larger than that of Nb, which results in a weaker scattering on phonons than that of the small-size inclusion. To avoid confusion, we have added some in-depth discussion to the manuscript.

Supplementary Figure 5. EDS mapping of $0.1\text{Nb}/\text{Mg}_3\text{Sb}_{1.5}\text{Bi}_{0.49}\text{Te}_{0.01}$ sintered at 1073 K on the fractured surface.

Supplementary Figure 15. EDS mapping of $0.1\text{Ta}/\text{Mg}_3\text{Sb}_{1.5}\text{Bi}_{0.49}\text{Te}_{0.01}$ sintered at 1073 K on the fractured surface.

Revision:

However, due to the higher κ ($57.5 \text{ W m}^{-1} \text{ K}^{-1}$) as well as weaker scattering originating from the larger size of Ta inclusions, the effect of Ta inclusions on reducing κ is limited (Supplementary Fig. 18).

5. I would suggest a predicted efficiency plot in Figure 6d.

Response:

Thank you for your suggestion. We have simulated the theoretical conversion efficiency by the COMSOL Multiphysics software and added the results to **Figure 6d** and SI.

By comparing the predicted (**Figure R9**) and measured parameters (**Supplementary Fig. 19**), it can be identified that the measured output power is slightly lower than the

predicted value and the heat flow is higher. The difference in output power mainly stems from the contact resistance. The excessive heat flow results from heat leakage due to technical limitations. The heating table area (10*10 mm) of the Mini-PEM device is larger than the sample size (4.5*5 mm), leading to heat being more easily radiated from the hot side to the cold side. As a result, the Q_c measured by the heat flow meter on the cold side is larger than the Q_c actually flowing through the sample. The predicted efficiency of the single-leg device is 18%. It is believed that the measured efficiency is expected to be improved by further optimizing the contact layer and controlling the influence of thermal radiation in the measurement.

Figure R9. The predicted (a) voltage, (b) output power (c) output heat and (d) efficiency as a function of the current under different hot-side temperatures for the single leg based on the zT value of 0.1Nb/Mg₃Sb_{1.5}Bi_{0.49}Te_{0.01}. The cold-side temperature is 300 K.

Revision:

The theoretical efficiency of the single-leg device reaches 18% as shown in Supplementary Fig. 22, but both the measured output power and heat flow deviate from the predicted value, indicating more efforts are needed to optimize the contact layer and control the thermal radiation.

Figure 6. (d) Comparison of the maximum conversion efficiency among the single leg in this work, other $\text{Mg}_3(\text{Sb,Bi})_2$ modules, and Bi_2Te_3 module under a series of temperature difference (ΔT)^{21,44-47}.

6. The authors used the phrases of "positive or negative effects" many times, these effects should be clearly defined.

Response:

We apologize for the unclear expression. We have corrected and polished the relevant statement in the manuscript.

Revision:

The results of Hall measurement suggest that the μ_H of the $x = 0.1$ sample shows a remarkable increase at low temperatures, whereas **this enhancement** is obviously weakened above 450 K.

The smaller grain size in the Nb-added samples sintered at 1073 K contributes to enhancing the grain boundary resistance, but the presence of Nb inclusion is able to cancel **the negative effect of grain boundary on electrical properties near room temperature**.

The fast transport channels for carriers formed by metallic inclusions cancel **the decrease in mobility caused by grain boundary scattering**, making the acoustic phonon scattering dominant.

Reviewer #3 (Remarks to the Author):

In this work, the authors reported the effects of additional metallic elements on thermoelectric properties of $\text{Mg}_3(\text{Sb,Bi})_2$ -based compounds. In previous studies, the higher electronic properties with Nb addition, larger grains were both already reported. However, the mobility value around room temperature is even higher than these reports and authors added some new insight into this improvement. Since Mg_3Sb_2 is a hot topic in the field of thermoelectrics and the results of this work are important for the development of thermoelectric devices, I can recommend publishing this work in nature communications. However, some additional explanations would be required before acceptance.

Response: Thanks for your positive and valuable comments.

1. Why Nb is chosen as the first example among all the possible elements? If the enhancement of the performance originated from lower grain boundary resistance and metallic inclusions, can we expect similar effects from most of the metals? Why Ta have less effective compared to Nb? Are there any indicators to identify the effective (beneficial) additives?

Response:

Thank you for your comments. Below are our respective responses to your questions.

Regarding the Nb as the first example.

As you mentioned, the previous study (*Adv. Funct. Mater.* **2021**, *31*, 2100258) has proved the higher electronic properties with Nb addition. However, the κ was also increased. In our previous work (*Adv. Mater.* **2023**, *35*, 2209119), it has been demonstrated that the κ can be significantly reduced by introducing defects through high-temperature sintering. Therefore, we would like to combine the advantages of Nb addition and high temperature sintering to achieve the optimization of TE performance in this work. According to the results, we have not only obtained an improvement of zT , but also found some phenomena that are different from the previous work, such as grain reduction.

#Regarding the expected similar effects from most of the metals.

The enhancement of performance can be expected if the chosen metal has similar properties to Nb. In the case of Nb addition, Nb does not enter the lattice, but acts as a metallic secondary phase. Moreover, the Nb inclusions bond well to the matrix after sintering. The interfaces between the inclusions and matrix have low electrical contact resistivity, which is confirmed by the investigation on the Nb/Mg₃SbBi layer (*Energy Environ. Sci.* **2022**, *15*, 3265 and *J. Inorg. Mater.* **2023**, *38*, 931).

In fact, the elements such as Nb, Mo and Cr are screened in Fu's work (*Energy Environ. Sci.* **2022**, *15*, 3265) by a high-throughput strategy to meet the requirements as diffusion barrier layer in Mg₃(Sb,Bi)₂ modules. These requirements include being almost chemically inert to TE materials, low electrical contact resistivity and good mechanical bonding, as shown in **Figure R11**. In recent work (*Adv. Energy Mater.* **2023**, 2301667), Mo is proven to be effective in improving zT as an additive. Therefore, it is expected that Cr or more elements have a similar effect.

[Redacted]

Figure R11. EDS mapping results showing the elemental distribution of different barrier layer candidates within the $\text{Mg}_{3.2}\text{Bi}_{0.996}\text{SbSe}_{0.004}$ matrix before and after thermal aging at 773 K for 360 h in a vacuum. (*Energy Environ. Sci.* **2022**, *15*, 3265)

#Regarding the less effectivity of Ta than Nb.

In our opinion, both Nb and Ta are beneficial in reducing the interfacial barrier. However, a trace of Nb_3Sb phase is observed at the interface after adding Nb, leading to the increased n_{H} . Different from the case of Nb addition, no Ta-Sb binary phase diagram is found. It may be difficult for Ta to form a Ta-rich secondary phase similar to Nb_3Sb . This may be the reason why the n_{H} of the Ta-added sample has a less increase than that of the Nb-added sample. Therefore, the σ of Nb-added samples increased more significantly.

Revision:

This may be because no Ta-rich secondary phases similar to Nb_3Sb were formed under the present experimental conditions. In fact, no Ta-Sb binary phase diagram was reported. That is why the Ta addition did not affect the carrier concentration as in the case of the Nb-added samples.

#Regarding the indicators to identify the effective (beneficial) additives.

Based on our experimental results, it is speculated that the effectiveness of additives can be judged by the following four indicators.

1. The large atomic size mismatch and the low solid solubility to make the additives not substituted at the lattice site.
2. The formation of the secondary phase with high conductivity or on the grain boundary, which is metal phase or metal-rich phase.
3. Intimate contact with low resistivity between the metallic inclusions and the matrix.
4. The good thermal stability of the secondary phase at the interface without diffusion or visible reaction.

2. Authors used Nb powder and Ta powder as additives. Is there any difference if the starting elements are different form such as smaller mesh or grains.

Response:

Thank you for your comments. In theory, the inclusions with smaller sizes can scatter

phonons more efficiently. However, the additives with smaller meshes or grains means that their specific surface areas will be enlarged, which can lead to high surface activity, thereby generating more opportunities to form agglomerations or get oxidized. In fact, we have tried to use Nb nano-powders, but indeed had no ideal results.

3. Page 3, line 93, "However, a slight decrease in S was observed at low temperatures in the $x=0.2$ sample". Do authors have any speculation of this change or is it caused by the measurement set up?

Response:

Thank you for your comments. The measurement set up is the same for all the results. Therefore, the decrease in S may be due to the increased n_H for $x=0.2$ sample. We have added an explanation of the decreased S to the paragraph discussing the n_H in the manuscript.

Revision:

Similarly, a more pronounced decrease in S is observed due to higher n_H in the $x = 0.2$ sample.

4. Page 3 line 111, could you please cite the previous report that authors are referring to?

Response:

Thank you for your suggestion. We have cited some references and revised the two sentences to make the expression more clear. Considering that the carrier scattering induced by point defects is enhanced with increasing temperature, the high-temperature electrical properties would be affected. However, in our work, the mobility is almost constant at high temperatures, but shows significant differences near room temperature. Therefore, the grain boundary effect is regarded as the dominant factor. The revisions are as follows.

Revision:

In fact, the changes in n_H and μ_H cannot be solely ascribed to the variable point defects (e.g. Mg vacancies²⁹ or doping atoms²⁷) as thought conventionally, because the carrier scattering induced by point defects is usually more intense in the high-temperature range, which significantly affects the electrical properties at high temperatures.

5. Based on their statement, I assume Nb is not substituting Mg site, and exists as the secondary phase at the grain boundary. Does that statement contradict the n_H changes depending on Nb content x ? (Line 118, Line 158) Considering the amount of Nb added ($x=0.1, 0.2$), isn't the carrier concentration change too small if it is depending on the Nb content x ?

Response:

Thank you for your comments. As you assumed, the Nb did not substitute the Mg site, which can be confirmed by the EDS results with no Nb signal in **Supplementary Fig. 4**.

Regarding the increased n_H with increasing Nb content, it can be considered as the decreased Sb/Mg ratio caused by the formation of the Nb_3Sb phase at the interface. Based on the results of the secondary phase of Nb_3Sb observed by HRTEM, we presume that the Sb atoms in Nb_3Sb can only come from the matrix. The EPMA results (**Figure R4**) can well support our statement that the Sb atoms in the matrix were consumed to form Nb_3Sb . For the samples sintered at 1073 K, the atomic ratio of Sb/Mg in the matrix of $x = 0.1$ sample is 0.595, slightly lower than 0.650 of $x = 0$ sample. Therefore, it is very possible to suppress the formation of Mg vacancies due to the decreased proportion of Sb. Therefore, when Mg vacancies as electron killers are reduced, the electron concentration, that is, the carrier concentration, would be increased, resulting in an enhancement in σ .

The more Nb is added, the more interfaces are introduced, and the more Sb atoms are consumed from the matrix, leading to a higher n_H . Due to the formed interfacial phase being only a trace, it can also explain why the amount of Nb added is so large but the increase in n_H is relatively slight.

Figure R4. Back-scattering images from the polished surface of $xNb/Mg_3Sb_{1.5}Bi_{0.49}Te_{0.01}$ ($x = 0$ and 0.01) sintered at 1073 K and corresponding point composition estimated by EPMA. The presence of Nb signals may be due to the residue of Nb inclusions during polishing or Nb inclusions whose size is less than the probe limit of 1 μm .

Revision:

Besides, due to the formation of the Nb_3Sb phase near the interface, a small amount of Sb atoms in the matrix would be inevitably consumed by Nb. As shown in

Supplementary Fig. 10, the results of electron probe microanalysis (EPMA) confirmed that the atomic ratio of Sb/Mg in the matrix of $x = 0.1$ sample is lower than that of the unadded sample. The decreased Sb/Mg ratio may suppress the formation of Mg vacancies, which is beneficial to increasing the n_H and hence the σ of $Mg_3(Sb,Bi)_2$.

6. Regarding the statement on Line 161, grain boundary growth is hindered by Nb inclusions, it seems different from the previous Nb added study on microstructure. (ref 22) Are there any differences in the condition or mechanisms?

Response:

Thank you for your comments. In fact, there is a difference between our work and this reference (*Adv. Funct. Mater.* **2021**, *31*, 2100258) on the observed phenomena. The grain size does decrease in the samples prepared by our process, which can be demonstrated by EBSD and EPMA of the polished surface. In our process, 8 hours of ball-milling time is chosen, longer than the 2 hours in the reference. And the samples are sintered at 1073 K and 50 MPa for 20 min in vacuum by spark plasma sintering in our work. In contrast, the samples are pressed by an induction-heating rapid hot press for 60 min at 873 K and 45 MPa under argon gas flow in the above reference.

As a result, the morphology of the secondary phase in the sintered samples is different. For the samples prepared by Luo et al. (*Adv. Funct. Mater.* **2021**, *31*, 2100258), Nb formed a wetting layer on the grain boundary, which was conducive to promoting grain growth. Besides, larger secondary phase of Nb, about tens of microns in size, was also observed as shown in **Figure R12**. For the bulks fabricated in our work, Nb exists as nanoparticles, which makes it easier to pin grain boundaries. This may be due to the longer milling time causing Nb to disperse in smaller particles and more uniform. In addition, it may also be due to the short sintering time so that it did not spread to form a wetting layer.

Figure R12. BSE image for a randomly selected large area and the corresponding EDS mapping of Nb_{0.1}-Mg₃Sb₂, showing the distribution of Nb, Mg, and Sb. (*Adv. Funct. Mater.* **2021**, *31*, 2100258)

7. What do author think about the mean free path of phonon and electron? Is there large enough difference between these two so that Nb nano inclusions (wide range of size of 10-1000nm) can selectively scatter only phonons without decreasing carrier mobility?

Response:

Thank you for your professional comments. The mean free path of the charge carrier (l_e) can be roughly calculated by the following equation (*J. Electron. Mater.* **2014**, *43*, 1733):

$$l_e = \frac{\hbar\sigma}{ne^2} (3\pi^2n)^{1/3}$$

where \hbar is Planck's constant, σ is the electrical conductivity, n is the carrier concentration, and e is the electric charge. The value of l_e is 10.0 nm at 300 K in this work. According to the work of Kanno et al. (*Adv. Funct. Mater.* **2021**, 2008469), the mean free path of phonons is 35 nm. Since the size of metallic inclusions is closer to the mean free path of phonons, phonons are more easily scattered than electrons. Actually, the difference between the mean free paths of phonons and electrons is not large enough. Therefore, the carriers would also be scattered, but weaker than phonons.

8. The energy filtering picture is not convincing. Is the difference in Seebeck coefficient large enough considering uncertainty and sample to sample difference etc.? How much improvement is coming from reduced grain boundary resistance and could it be analyzed separately from energy filtering effect? Further analysis would be required to use energy filtering story to explain the change in electrical properties.

Response:

Thank you for your comments.

#Regarding the effect of grain boundary resistance.

As shown in Fig. 1 and 2, after Nb adding, the grain size is decreased but the σ is improved at low temperatures. Therefore, the grain boundary resistance is actually increased instead of reduced. Considering the increased grain boundaries enhance the interface resistance, together with the increased number of interfaces generated by Nb inclusions, it can be inferred that the interfaces between Nb and matrix should own lower resistance than the grain boundaries. Based on this, it is believed that the improvement of electrical properties is due to the addition of metallic inclusions. There is a relevant discussion in the manuscript.

#Regarding the energy filtering effect.

Different from the generally mentioned energy-filtering effect via introducing barriers, the tailored energy-filtering effect we discussed in this work is caused by the reduced interfacial barrier when considering the introduced barriers by grain boundaries. The previous work (*Energy Environ. Sci.* **2009**, 2, 466) has shown that grain boundaries can negatively affect mobility, but also possibly play a positive role through energy filtering. As schematically shown in Figure R7, considering that the barrier introduced by grain boundaries is located at E_{b1} (green box), the electrons with negative Seebeck distribution and a part of electrons with positive Seebeck distribution are scattered by the interfacial barriers. After adding metallic inclusions, the interfacial barrier would be reduced from E_{b1} to E_{b2} (red box). In this case, more electrons with positive Seebeck distribution are able to pass through the barrier, thus increasing the Seebeck coefficient.

Consequently, it is believed that the addition of metallic inclusions and the reduced interfacial barriers have a favorable regulation of the existing energy-filtering effect. To avoid misunderstanding, we have revised the manuscript and replaced the expression of “tailored energy filtering effect” with “modified interfacial barrier”.

[Redacted]

Figure R7. Calculated normalized Seebeck distribution versus energy. Low energy electrons reduce the total Seebeck coefficient. (*Energy Environ. Sci.* **2009**, 2, 466).

#Regarding the difference in Seebeck coefficient.

As mentioned above, reducing the interfacial barrier is beneficial for increasing S . Meanwhile, the S would exhibit a decrease when n_H increases. Therefore, the S may be a synergistic result of the above effect and increased n_H . Considering the n_H of the samples in this work is increased, it is difficult to evaluate the actual increase of S . More importantly, the S remains almost unchanged when the σ increases substantially. Herein, what we would like to emphasize is the decoupling of the σ and S due to the reduction of interfacial barriers by adding metallic inclusions.

Revision:

Meanwhile, the S could be obviously enhanced in the $Mg_3(Sb,Bi)_2$ with Ta addition, indicating that these built-in metallic nano-inclusions effectively modified the interfacial barriers and enhanced the contribution from high-energy electrons to transport properties.

By incorporating Ta inclusions, the interfacial barrier is reduced from E_{b1} to E_{b2} , allowing more high-energy carriers to pass through. The contribution of high-energy carriers to transport properties is enhanced, accounting for a higher S . In the case for Nb-added samples, this modified interfacial barrier turns out to optimize σ without sacrificing S , thereby leading to enhanced PF.

REVIEWER COMMENTS

Reviewer #1 (Remarks to the Author):

I have read through the responses from authors, the revision and clarification seem quite well prepared, I am basically satisfied with the explanation and experimental input in this revised version.

However, one remaining concern lie in that the readers may not be convinced with the claim of "fast electron transport channels provided by the embedded metallic inclusions", considering the volume ratio of metallic inclusions (Nb) in the $Mg_3(Sb,Bi)_2$ matrix is less than about 1%, how could this trivial amount "channels" give rise to relative 30% increase in the electrical conductivity? the authors don't have a quantitative model to illustrate this enhancement. On the contrary, i believe the concept of "modified interfacial effect" might be the key. To avoid misleading, i suggest the author to weaken the claim of "fast electron transport channels" in the paper.

Reviewer #2 (Remarks to the Author):

I have gone through the comments raised from the other reviewers and mine, and the corresponding responses as well. There does not appear to be a problem with publishing it.

Reviewer #3 (Remarks to the Author):

Thanks to the detailed explanation and some additional data, analysis shown in both SI and the main text. Their clarification of the experimental details on the experimental procedure and the validity of their claim should be helpful to understand the results of this work. The added explanations are also essential to improve accuracy and their statements are strengthened with the additional data. As the revised paper is polished and more understandable, I would recommend accepting this article to Nature Communications.

Sep. 28th, 2023

Dear Respected Editor and reviewers,

Thank you very much for your time and efforts in handling our manuscript (Research Article, No. NCOMMS-23-30931A).

We are pleased that all the reviewers agreed with our replies and revisions. We greatly appreciate the further comments and suggestions from reviewer#1. We have revised our manuscript again, especially regarding his/her major concern about the concept of "fast electron transport channels". We hope our following revisions are satisfactory and this revised manuscript will address the concerns of the reviewer#1 and meet the requirements of your esteemed journal *Nature Communications*.

Answers to reviewers:

Reviewer #1:

I have read through the responses from authors, the revision and clarification seem quite well prepared, I am basically satisfied with the explanation and experimental input in this revised version.

However, one remaining concern lie in that the readers may not be convinced with the claim of "fast electron transport channels provided by the embedded metallic inclusions", considering the volume ratio of metallic inclusions (Nb) in the $Mg_3(Sb,Bi)_2$ matrix is less than about 1%. how could this trivial amount "channels" give rise to relative 30% increase in the electrical conductivity? The authors don't have a quantitative model to illustrate this enhancement. On the contrary, I believe the concept of "modified interfacial effect" might be the key. To avoid misleading, I suggest the author to weaken the claim of "fast electron transport channels" in the paper.

Response: We appreciate your recognition of our revisions. Thank you very much for your rigorous and professional review. We apologize for the misnomer in the claim of "fast electron transport channels provided by the embedded metallic inclusions". Our intended meaning was that charge carriers can easily transport through the lower potential barriers induced by these inclusions, as compared to grain boundaries with high potential barriers. Therefore, it would be more accurate to replace 'fast' with 'easy' and 'channel' with 'gate'. It should be the inappropriate metaphor that led to the misunderstanding.

Indeed, the word "channels" is commonly associated with the interconnected network as the path for charge carrier transport. We guess that you may have thought of it that way. In this scenario, the interconnected network was formed through the percolation effect, making it indeed difficult to achieve with such a small amount of 1 vol.%.

Based on your suggestion, we have weakened the claim of "fast electron transport

channels" and emphasized the "modified interfacial effect". All the related terminology in the text has been modified to "facilitate carrier transport", "the modified interfacial effect on carrier transport", or "reduce the interfacial barriers and promote carrier transport". And we have added arrows about the grain boundary scattering into the schematic illustration (Fig. 3g) to better exhibit the carrier transport behaviors. Please refer to the revised manuscript. We hope the reviewer will be satisfied with the changes to the manuscript.

Revision:

The reduced interfacial barriers help to improve the carrier transport at low and high temperatures.

It was inferred that the embedded Nb inclusions at grain boundaries can reduce the interfacial barriers, so that the carriers can pass easily through the interfaces.

Therefore, the embedded Nb inclusions at grain boundaries should help to reduce the interfacial barriers and weaken the grain boundary scattering, which promotes carrier transport, as schematically shown in Fig. 3g.

The modified interfacial effect by metallic inclusions cancels the decrease in mobility caused by grain boundary scattering, making the acoustic phonon scattering dominant. These composition-transition interfaces with high conductivity further support the modulation of interfacial barriers by Nb inclusions.

The incorporation of Nb and Ta nano-inclusions into the $Mg_3(Sb,Bi)_2$ matrix helps to modify the interfacial barriers, producing similar favorable modulation results.

The reduced interfacial barriers significantly increased the electrical conductivity in the low-temperature range.

Fig. 3 Microstructures and interface analysis. (g) Schematic illustration of different carrier transport behaviors at grain boundaries and metallic inclusions.

Reviewer #2:

I have gone through the comments raised from the other reviewers and mine, and the corresponding responses as well. There does not appear to be a problem with publishing it.

Response: Thank you for your recognition. We are glad to address your concerns for

this work.

Reviewer #3:

Thanks to the detailed explanation and some additional data, analysis shown in both SI and the main text, their clarification of the experimental details on the experimental procedure and the validity of their claim should be helpful to understand the results of this work, The added explanations are also essential to improve accuracy and their statements are strengthened with the additional data. As the revised paper is polished and more understandable, I would recommend accepting this article to Nature Communications.

Response: Thank you for your recognition. We are glad to address your concerns for this work.

REVIEWERS' COMMENTS

Reviewer #1 (Remarks to the Author):

I believe the authors have basically addressed my concerns, i would recomment to accept for publication.

Oct. 23th, 2023

Dear Respected Editor and reviewers,

Thank you very much for your time and efforts in handling our manuscript (Research Article, No. NCOMMS-23-30931B).

We are pleased that the reviewer#1 agreed with our replies and revisions and our manuscript has met the requirements of your esteemed journal *Nature Communications*. According to the Author Checklist, we further checked the manuscript and other files. And we revised the end of the abstract following the formatting instructions.

Answers to reviewers:

Reviewer #1:

I believe the authors have basically addressed my concerns, i would recommend to accept for publication.

Response: Thank you for your recognition. We are glad to address your concerns for this work.